# Federated Continual Learning via Orchestrating Multi-Scale Expertise

**Xiaoyang Yi**[1,3,4]    **Yang Liu**[2]    **Binhan Yang**[2]    **Jian Zhang**[1,2,3,4,*]

[1]College of Cryptology and Cyber Science, Nankai University, China

[2]College of Computer Science, Nankai University, China

[3]Tianjin Key Laboratory of Network and Data Security Technology, Tianjin, China

[4]Key Laboratory of Data and Intelligent System Security, Ministry of Education, Tianjin, China

{xiaoyangyi, liuyang99. yangbinhan}@mail.nankai.edu.cn

zhang.jian@nankai.edu.cn

## Abstract

Federated continual learning (FCL) aims to maintain the model's performance on old tasks (i.e., stability) while enhancing its ability to acquire knowledge from current tasks (i.e., plasticity). With the development of pre-trained models (PTMs), fine-tuning PTMs on clients has become a promising approach to leveraging their extensive knowledge in FCL. In this paper, we propose MultiFCL, a novel FCL framework that fine-tunes PTMs to adapt to FCL while preserving their strong generalization capabilities. Specifically, to ensure the stability, MultiFCL introduces lightweight adapters for task adaption, which are subsequently frozen to prevent catastrophic forgetting. Moreover, by utilizing the semantic features of old tasks, MultiFCL performs multi-modal initialization of new task class prototypes. To enhance the plasticity, MultiFCL employs a multi-expert training mechanism that integrates multi-scale feature learning with multi-teacher dynamic self-distillation. Through intra-client and inter-client expert communication, MultiFCL facilitates cross-task and cross-client knowledge fusion. Experimental results demonstrate that MultiFCL achieves state-of-the-art performance across multiple datasets and settings, showcasing its effectiveness in FCL scenarios.

## 1 Introduction

Federated learning (FL) is a distributed machine learning paradigm that allows clients to collaboratively train models without sharing sensitive data (McMahan et al., 2017). In recent years, FL has gained significant attention across various domains, such as healthcare (Yang et al., 2021; Ogier du Terrail et al., 2022; Chen et al., 2024b; Xie et al., 2024) and the internet of things (IoT) (Nguyen et al., 2023; Cui et al., 2022; Yu et al., 2024a; Jia et al., 2024). Traditional FL approaches typically assume that client data distributions do not change over time. However, this assumption is unrealistic in practical applications, as client data usually evolves continuously (Ma et al., 2022; Lu et al., 2024). For example, healthcare models must adapt to emerging diseases and their variants (Babakniya et al., 2023), while IoT models need to keep pace with changes in user behaviors (Zhang et al., 2024). When there are such shifts in data distributions, directly retraining existing models on new data can lead to catastrophic forgetting (Smith et al., 2023b; Li et al., 2024; Yu et al., 2024b), which significantly impairs the model's performance on previously learned tasks.

To address this, federated continual learning (FCL) integrates continual learning (CL) techniques into FL (Yoon et al., 2021), aiming to seek a balance between stability (i.e., maintaining performance on

---

[*]Corresponding Author

39th Conference on Neural Information Processing Systems (NeurIPS 2025).

old tasks) and plasticity (i.e., acquiring knowledge from new tasks). Unlike traditional CL methods that rely on extensive data retention to perform experience replay, many FCL approaches instead opt for generative replay (Qi et al., 2023; Wuerkaixi et al., 2024; Tran et al., 2024) or model decomposition (Bakman et al., 2024), due to privacy concerns and resource limitations. These approaches train models from scratch, trying to continuously absorb knowledge from new tasks while reviewing knowledge from previous ones (Zhou et al., 2024b). However, the heterogeneity among client data presents challenges in maintaining both stability and plasticity during the process of training a model from scratch. Fortunately, the emergence of pre-trained models (PTMs) has offered a promising alternative, owing to their remarkable generalization capabilities (Dosovitskiy et al., 2021; Radford et al., 2021). Trained on large-scale datasets, PTMs are better equipped to acquire optimal knowledge compared to incrementally trained model, thereby significantly alleviating the learning burden (Zhou et al., 2024c).

Although PTMs exhibit strong generalization capabilities, how to fine-tune them for downstream tasks is still a dilemma: fully fine-tuning risks diminishing their generalizable features, while freezing the backbone can prevent the integration of downstream information (Zhou et al., 2024a). To this end, some FCL approaches incorporate trainable prompt parameters into PTMs, training prompts locally on clients and updating the global prompt pool on the server by aggregating local prompt pools (Halbe et al., 2024; Yu et al., 2024c). A query mechanism is then employed to dynamically select a subset of prompts from the global prompt pool, guiding PTMs in adapting to different data distributions (Wang et al., 2022; Smith et al., 2023a). By keeping the pre-trained weights frozen during this process, these prompt-based approaches effectively preserve the generalizability of PTMs while allowing task-specific adaptions. However, prompt selection often focuses on specific subsets, limiting the adaptability to broader data distributions (Moon et al., 2023). Moreover, as new data accumulates, the prompt pool size expands, leading to increased communication overhead, whereas restricting the prompt pool size constrains its representational capacity (Piao et al., 2024).

Inspired by these insights, we propose a novel FCL framework, MultiFCL, which effectively fine-tunes PTMs while preserving their strong generalization capabilities to adapt to FCL. Specifically, to maintain the stability of PTMs, MultiFCL employs lightweight adapters to quickly adapt PTMs to downstream image classification tasks, and then freezes them to prevent catastrophic forgetting. Furthermore, MultiFCL leverages semantic features from old tasks to perform multi-modal initialization of new class prototypes, thereby incorporating prior knowledge. To enhance the plasticity of PTMs, MultiFCL introduces a multi-expert training mechanism, which combines multi-scale feature learning and multi-teacher dynamic self-distillation to strengthen feature representation for current tasks and achieve effective knowledge integration. Through both intra-client and inter-client expert communication, MultiFCL facilitates cross-task and cross-client knowledge fusion, addressing the dynamic and heterogeneous data distribution challenges inherent in FCL.

Our contributions are as follows:

- We propose MultiFCL, a novel FCL framework that preserves the strong generalization capabilities of PTMs while fine-tuning them, enabling PTMs to maintain both stability and plasticity.

- To ensure stability, we employ lightweight adapters for task-specific fine-tuning and utilize multi-modal initialization of new task class prototypes based on semantic features from old tasks.

- To enhance plasticity, we design a multi-expert training method that combines multi-scale feature learning and multi-teacher dynamic self-distillation, achieving effective knowledge fusion.

- Experimental results demonstrate that MultiFCL achieves state-of-the-art performance across multiple datasets and configurations, highlighting its effectiveness and the robustness of PTMs.

## 2 Related Work

FCL can be broadly categorized into task-incremental learning (TIL) and class-incremental learning (CIL). The former focuses on learning new tasks sequentially, while the latter involves learning classes in each task incrementally. Specifically, Federated TIL was first introduced in FedWeIT (Yoon et al., 2021). To prevent forgetting old tasks while learning new ones, FedWeIT decomposes network weights into global parameters and sparse task-specific parameters, each client selectively transfers knowledge from others by their task-specific parameters. CFeD (Ma et al., 2022) employs continual distillation using an unlabeled proxy dataset on both clients and the server, assigning different learning

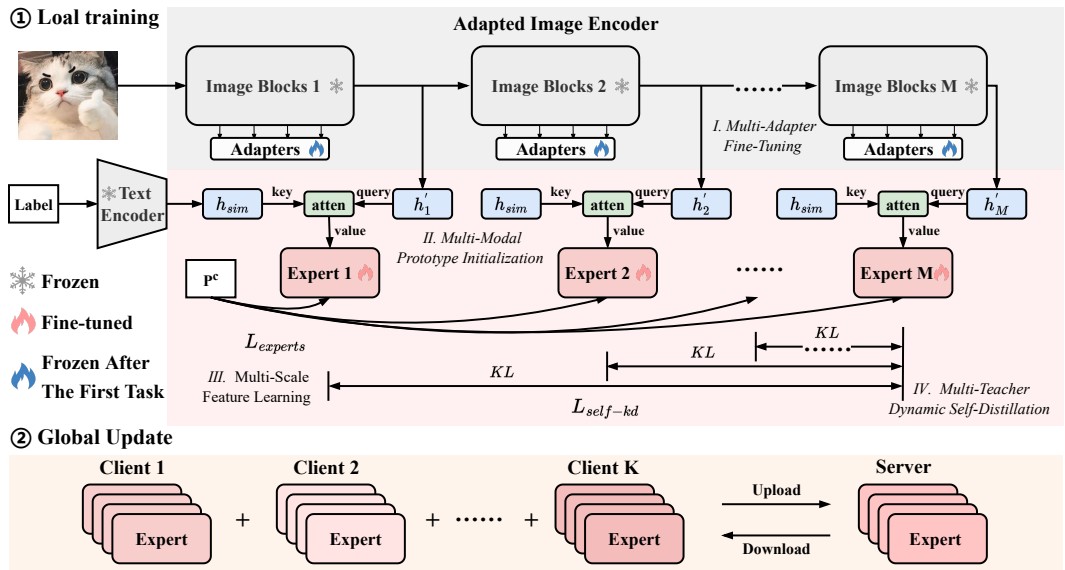

Figure 1: The overall framework of MultiFCL. MultiFCL employs adapters to fine-tune the PTM and leverages the semantic features of old tasks to initialize new class prototypes. Then, MultiFCL establishes multiple experts, employing feature learning loss $\mathcal{L}_{\text{experts}}$ and the multi-teacher dynamic self-distillation $\mathcal{L}_{\text{self-kd}}$ to transfer knowledge to the final expert.

objectives to clients for learning new tasks and revisiting old ones. AF-FCL (Wuerkaixi et al., 2024) selectively leverages prior knowledge in the federated network and uses a probabilistic framework based on normalized flow models to quantify the reliability of this prior knowledge. FOT (Bakman et al., 2024) extracts global input subspaces for old tasks at each layer and modifies the aggregated updates for new tasks to ensure orthogonality with these global subspaces. Powder (Piao et al., 2024) introduces a novel prompt generation and aggregation method to effectively facilitate knowledge transfer encapsulated in prompts across sequential tasks and clients. FedMGP (Yu et al., 2024c) employs coarse-grained global prompts for efficient shared knowledge transfer and fine-grained local prompts for personalized learning, mitigating spatial and temporal forgetting.

Class-incremental FCL, also known as federated class-incremental learning (FCIL), was first introduced by GLFC (Dong et al., 2022). To mitigate forgetting of old classes while learning new ones, GLFC proposes a class-aware gradient compensation loss and a class-semantic relation distillation loss to balance old-class retention. FedCIL (Qi et al., 2023) employs ACGAN (Odena et al., 2017) to synthesize data from old distributions for replay, alleviating catastrophic forgetting. Similarly, MFCL (Babakniya et al., 2023) uses a generative model to synthesize samples from old data distributions and trains on the server using a data-free method. FedET (Liu et al., 2023) introduces a small Enhancer module to absorb and communicate new knowledge and proposes an enhancer distillation method to address the imbalance between old and new knowledge. LANDER (Tran et al., 2024) uses label text embeddings as anchors to constrain the feature embeddings of corresponding training samples, generating meaningful samples for replay. FedCBC (Yu et al., 2024b) constructs a class-specific binary classifier for each class, extracting prior knowledge from the global model into multiple local models for selective knowledge fusion. LGA (Dong et al., 2024) introduces a class-balanced gradient adaptive compensation loss and a class-gradient-induced semantic distillation loss to balance the heterogeneous forgetting rates of old classes.

## 3 Method

The MultiFCL framework is shown in Figure 1. Specifically, to adapt the powerful PTM to downstream image classification tasks, MultiFCL introduces adapters into the PTM for task-specific fine-tuning (Section 3.1). These adapters remain frozen during the training process of subsequent tasks, effectively mitigating catastrophic forgetting. By integrating semantic features from old tasks with those from new tasks, MultiFCL performs multi-modal initialization of new task prototypes (Sec-

tion 3.2), which enhances the representation of new class features while preserving old knowledge. During the training process, multi-scale features and new task prototypes are used for multi-scale feature learning to ensure the plasticity (Section 3.3). Additionally, MultiFCL employs multi-teacher self-distillation to transfer knowledge to the final expert, and dynamically adjusts the distillation loss using uncertainty-weighted perception in the knowledge transfer process (Section 3.4).

## 3.1 Multi-Adapter Fine-Tuning

In FCL, each client $k$ ($1 \leq k \leq K$) trains a local model using its local dataset $D_k^{\mathcal{T}} = \{(x_i, y_i)\}_{i=1}^{n_k^{\mathcal{T}}}$ to sequentially deal with tasks $\{\mathcal{T}_1, \mathcal{T}_2, \cdots, \mathcal{T}_{\mathcal{T}}\}$, where $n_k^{\mathcal{T}}$ is the dataset size for task $\mathcal{T}$ on client $k$. For the same task $\mathcal{T}$, data distributions across clients are non-independent and identically distributed (Non-IID). Meanwhile, within the same client $k$, the class sets across different tasks are disjoint. The overall optimization objective for clients is two fold: (1) minimizing the loss function $L(\cdot)$ on the current task's dataset to achieve high performance (i.e., plasticity), (2) and retaining knowledge from previously learned tasks' datasets to mitigate catastrophic forgetting (i.e., stability).

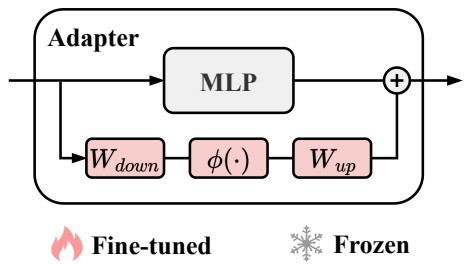

Figure 2: The structure of a adapter in the PTM.

To bridge the domain knowledge gap between the PTM and downstream image classification tasks, we introduce trainable adapter parameters $w_k$ into the frozen PTM, enabling adaptation to FCL tasks for each client $k$. Adapter introduces small neural network modules independently into various layers of the model, separate from its main structure. Only these adapter parameters are updated during fine-tuning. In contrast, LoRA (Hu et al., 2022) introduces low-rank matrices into the model's weight matrices. These low-rank matrices act as modifications applied to the original weight matrices, adjusting them during computation. Specifically, we employ a transformer-based PTM (e.g., CLIP (Radford et al., 2021)) as the backbone model, which consists of multiple repeated blocks with several layers. Adapters are inserted into each layer of the image encoder to facilitate fine-tuning. As illustrated in Figure 2, each adapter is a bottleneck module, which is composed of a down-sampling layer $W_{down}$ for reducing feature dimensions, a nonlinear activation function $\phi(\cdot)$ (e.g., ReLU (Glorot et al., 2011)), and an up-sampling layer $W_{up}$ for restoring the original dimensions (Chen et al., 2024a). This process can be formulated as:

$$h^{'} = h + \phi(hW_{down})W_{up} \tag{1}$$

where $h$ and $h^{'}$ are the input and output feature representations through an adapter, respectively.

Since the local dataset of client $k$ constantly evolves with changing tasks, training adapters for each task can easily lead to catastrophic forgetting. To fully preserve the generalization ability of the PTM, we fine-tune adapters only on the first task and then freeze them to prevent excessive updates, as follows:

$$f'(x) = L(f(x), D_k^1, w_k) \tag{2}$$

Here, $f(\cdot)$ represents the original PTM, $f'(\cdot)$ is the PTM with adapters, and $D_k^1$ is the dataset for the first task of client $k$. By fine-tuning with adapters on the first task, the PTM can rapidly adapt to downstream image classification tasks.

During each communication round $t$ of the first task training, each client uploads its locally trained adapters to the server for aggregation, enabling the model to acquire more comprehensive knowledge as follows:

$$w^t = \sum_{k=1}^{K} \frac{\left|X_k^1\right|}{\sum_{k=1}^{K} \left|X_k^1\right|} w_k^t \tag{3}$$

where $w^t$ denotes the global adapters, and $\left|X_k^1\right|$ is the total number of samples owned by client $k$ for the first task. Once the first task is completed, all adapters will be frozen and no longer updated during model training, preventing performance from degrading due to excessive fine-tuning.

## 3.2 Multi-Modal Prototype Initialization

Adapter fine-tuning enables the PTM to quickly adapt to downstream image classification tasks, while for classification tasks, a classifier is also needed to ensure its generalizability. Since the adapter is only trained in the first task, it focuses more on the knowledge related to that task and may overlook some irrelevant high-level features (Zhou et al., 2024b). Therefore, instead of using a conventional linear classifier for classification, we use prototypes as weights of the classifier, performing classification by measuring the distance between class prototypes and sample features. Specifically, we continuously update the optimal class prototype $\mathbf{p}^c$ ($c \in \mathcal{T}$) through training, as follows:

$$\mathbf{p}^c_{(0)} = \mathbf{p}^c_{\text{init}} \tag{4}$$

$$\mathbf{p}^c_{(j)} = \frac{\mathbf{p}^c_{(j-1)} + f'(\mathbf{x}^c_j)}{2}, \quad \forall j \in \{1, 2, \ldots, n^{\mathcal{T}}_c\} \tag{5}$$

where $\mathbf{p}^c_{\text{init}}$ is the initialization prototype of class $c$, $\mathbf{p}^c_{(j)}$ is the current prototype from class $c$ to sample $\mathbf{x}^c_j$, and $n^{\mathcal{T}}_c$ is the number of class $c$ in the task $\mathcal{T}$.

A common initialization method is random initialization, which makes the training of the classifier independent learning around the task. For subsequent tasks, it may lead to the classifier of the new task lacking knowledge from old tasks, resulting in lower discriminability (Kurniawan et al., 2024). To address this, the bias-adjustment heuristic (Kahneman et al., 1982) is proposed, which uses available related knowledge to estimate unknowns. Therefore, we introduce relevant semantic features from old tasks as available knowledge to estimate the new class prototypes, and use multi-modal fusion to form their initialized prototypes.

In order to fuse semantic features with sample features of the new class for initializing the new class prototype, we first calculate the similarity between the feature of the first sample of the new class and the semantic features $C^{\mathcal{T}_{\text{old}}}$ of the old task labels to establish the basic relationship assumption between the old and new classes, as follows:

$$\text{sim}\left(f'(x^c_1), f_{text}(C^{\mathcal{T}_{\text{old}}})\right) = \frac{f'(x^c_1) \cdot f_{\text{text}}(C^{\mathcal{T}_{\text{old}}})}{\|f'(x^c_1)\|\|f_{\text{text}}(C^{\mathcal{T}_{\text{old}}})\|} \tag{6}$$

where $\text{sim}(\cdot)$ is a similarity calculation function, and $f_{text}(\cdot)$ is the text encoder of the PTM.

Subsequently, the most similar label semantic features are selected and merged with the sample features of the new class through a cross-attention mechanism to initialize the new class prototype, thereby incorporating prior implicit bias as follows:

$$\mathbf{p}^c_{\text{init}} = \text{Attention}\left(f'(x^c_1), f_{text}(c^{\mathcal{T}_{\text{old}}}_{\text{sim}}), \mathbf{p}^c_{\text{old}}\right) \tag{7}$$

where $c^{\mathcal{T}_{\text{old}}}_{\text{sim}}$ and $\mathbf{p}^c_{\text{old}}$ is the semantic feature and the class prototype of the old task that is most similar to the current class, respectively. $\text{Attention}(\cdot)$ is the cross-attention. The initialized prototypes obtained in this way can enhance the feature representation of the new class while preventing the forgetting of old class knowledge, thereby ensuring the stability of the model.

## 3.3 Multi-Scale Feature Learning

While ensuring the stability of the model, it is also crucial to maintain its plasticity, which refers to the model's ability to acquire knowledge from the current task. Research (Jung et al., 2023) has shown that multi-scale features can help the model distinguish between different samples by providing multi-level information, ranging from shallow to deep knowledge. Therefore, we divide the PTM into $M$ modules, each consisting of multiple blocks, which can be treated as experts with different levels of knowledge.

Specifically, the feature output from the last layer of each module serves as the input for the next module. Additionally, the features output by these modules will be used to initialize and update the new task class prototypes, as described in Section 3.2. By utilizing these prototypes, the output features at different scales from each module are continuously trained to move closer to their corresponding class centers, allowing the experts to effectively consolidate the class knowledge

of each new task. Subsequently, the experts optimize the acquired knowledge through mutual communication, with the optimization objective as:

$$\mathcal{L}_{\text{experts}} = \sum_{m=1}^{M} \left( -\sum_{x_i \in \mathcal{T}} y_i \log p_{x_i,m} \right) \tag{8}$$

where $M$ is the number of experts, and $p_{x_i,m}$ represents the prediction probability that sample $x_i$ belongs to label $y_i$ in expert $m$. Through multi-scale feature learning, the multi-level experts in clients are able to make decisions from different perspectives on the same sample, thus capturing the diversity and complexity of the sample more comprehensively. As the data distribution continues to evolve, clients become more flexible in adapting to new tasks.

In addition to the collaboration of experts within each client to jointly classify samples, coordination and communication between clients are also necessary. To achieve this, during each communication round $t$, each client uploads the class prototypes obtained after local training to the server for aggregation as follows:

$$\mathbf{p}^{c,t} = \sum_{k=1}^{K} \frac{\left| X_k^{c,t} \right|}{\sum_{k=1}^{K} \left| X_k^{c,t} \right|} \sum_{m}^{M} \mathbf{p}_{k,m}^{c,t} \tag{9}$$

where $\mathbf{p}^{c,t}$ represents the global class prototype, $\left| X_k^{c,t} \right|$ denotes the total samples of class $c$ in client $k$, and $\mathbf{p}_{k,m}^{c,t}$ is the class prototype in expert $m$ of client $k$. As a result, the server collects the multi-scale prototypes from different clients to form global class prototypes, which are then sent back to the clients as classifier weights for each module in the next round of training. By continuously sharing and updating class prototypes, clients are able to better learn representative multi-scale knowledge during the federated learning process. This provides more precise task guidance for subsequent feature learning.

### 3.4 Multi-Teacher Dynamic Self-Distillation

Multi-scale feature learning enables better predictions by allowing different experts to make judgments on the samples from different levels of knowledge (Aljundi et al., 2017). However, using multiple experts to generate predictions introduces additional computational overhead, which becomes particularly redundant during testing (Yan et al., 2024). Therefore, we propose a novel self-distillation method that transfers knowledge from multi-level experts to the final high-level expert. Among them, the former experts act as teachers, while the latter serves as the learner to integrate the knowledge from all the experts, as follows:

$$\mathcal{L}_{\text{self-kd}} = \sum_{m=1}^{M-1} \text{KL} \left( \sigma(x_{i,m}) \,\|\, \sigma(x_{i,\text{final}}) \right) \tag{10}$$

where $\text{KL}(\cdot)$ is the Kullback-Leibler divergence function, and $\sigma(\cdot)$ represents the softmax function. Moreover, $x_{i,m}$ and $x_{i,\text{final}}$ denote the output of expert $m$ and the final expert, respectively.

It calculates the distance between the output of each expert and the output of the final expert, thereby consolidating the knowledge from multi-level experts into the final expert and reducing the number of experts that need to be computed during testing. During training, the distillation loss encourages the final expert to learn from the knowledge of all the experts, including the fine-grained features from lower-level experts and the abstract decisions from higher-level experts. In this way, the final expert is able to independently perform the task while avoiding excessive reliance on the features from the final layer, thereby obtaining more comprehensive decision-making capabilities.

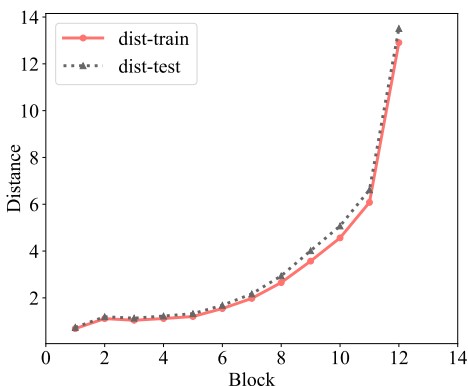

Figure 3: The distance between feature outputs at each block.

During self-distillation, the output quality of different experts may vary. Figure 3 shows the distance between the feature outputs at each block of the PTM, with a significant difference between deep and shallow features. Simply adding the distance between each expert's output and that of the final expert does not fully reflect each expert's contribution to the task, as it tends to emphasize shallow knowledge. Therefore, we propose an uncertainty-weighting method to dynamically adjust the importance of each expert in the self-distillation process.

Specifically, the output quality of each expert is measured not only by the distance between its output and the final expert's output but also by considering the uncertainty of its output features. To quantify uncertainty, we use the entropy of the output from each expert. A higher entropy indicates that the expert's output is more ambiguous or uncertain, and its contribution should be correspondingly reduced. Conversely, an expert with lower entropy represents more certain outputs, which can provide more stable knowledge, and thus should be assigned a higher weight, as follows:

$$\lambda_{\text{expert}} = \frac{1}{\text{KL}\left(\sigma(x_{i,\text{expert}}) \parallel \sigma(x_{i,\text{final}})\right) + \epsilon} \tag{11}$$

where $\epsilon$ is a small positive constant used to prevent the denominator from becoming zero and to avoid the weights from growing infinitely large.

To ensure fairness among the weights of different experts, we normalize these weights as follows:

$$\lambda'_{\text{expert}} = \frac{\lambda_{\text{expert}}}{\sum_{m=1}^{M} \lambda_m} \tag{12}$$

By suppressing the weights of shallow experts with high uncertainty and enhancing the weights of deep experts with low uncertainty, the knowledge from experts at various layers can be more effectively integrated. It not only prevents excessive interference from shallow knowledge during the self-distillation process, but also ensures that deep knowledge is fully utilized during learning, thereby enhancing the plasticity of the model.

Ultimately, the overall loss with multi-scale feature learning and multi-teacher dynamic self-distillation is as follows:

$$\mathcal{L} = \mathcal{L}_{\text{experts}} + \lambda'_{\text{expert}} \cdot \mathcal{L}_{\text{self-kd}} \tag{13}$$

Through cross-task and cross-client multi-modal knowledge fusion, MultiFCL effectively handles changes in data distribution in the FCL setting, preventing catastrophic forgetting while maintaining the powerful capabilities of the PTM.

## 4 Experiments

### 4.1 Experiment Setting

We evaluate CIL and TIL experiments using CLIP (Radford et al., 2021) as our base model, with the image encoder being ViT (Dosovitskiy et al., 2021). In addition, we use CIFAR100 (Krizhevsky & Hinton, 2009), TinyImageNet (mnmoustafa & Ali, 2017), ImageNet-R (Hendrycks et al., 2021), and CUB-200-2011 (Wah et al., 2011), which are not learned by CLIP during pre-training. We divide all datasets into 5 and 10 tasks for incremental learning, using accuracy in the last task (Last), average accuracy in all tasks (Avg.), and the forgetting rate $\mathcal{F}$ as the metric for evaluation, where $\mathcal{F}$ is defined as the difference in accuracy between the first task and the final task. We present partial results for CIL on CIFAR100, with the remaining experiments in Appendix.A. Following the setting of existing FCL methods, we employ two different data partition strategies, including distribution-based partitioning and quality-based partitioning, where the former fixes the number of samples per class and specifies the number of classes owned by the client through $\alpha$, while the latter uses $\beta$ to specify the degree of data heterogeneity based on the Dirichlet distribution. Each partition is further subdivided into three scenarios, with specific settings provided in Appendix.B.

To demonstrate the capabilities of PTMs, we compare ViT and CLIP with FL implementations of several FCL methods such as MFCL (Babakniya et al., 2023), LANDER (Tran et al., 2024) and FedCBC (Yu et al., 2024b). Furthermore, to validate the effectiveness of MultiFCL, we compare it with PTM-based FCL methods such as FedMGP (Yu et al., 2024c) and PiLoRA (Guo et al., 2024), which also use ViT. We also replace the base models of MFCL and PiLoRA with CLIP to maintain

Table 1: Main results of performance comparison on CIFAR100 with 10 tasks in distribution-based partitioning, where -△ means using CLIP as base model. The best results are shown in bold.

| | Methods | $\beta = 0.5$ | | | $\beta = 0.1$ | | | $\beta = 0.05$ | | |
|---|---|---|---|---|---|---|---|---|---|---|
| | | Last | Avg. | $\mathcal{F}$ | Last | Avg. | $\mathcal{F}$ | Last | Avg. | $\mathcal{F}$ |
| ViT | ICLR 2021 | 65.0 | 68.2 | 22.9 | 63.3 | 66.7 | 23.7 | 60.3 | 64.4 | 24.2 |
| CLIP | ICML 2021 | 65.0 | 68.2 | 25.6 | 63.3 | 66.7 | 27.8 | 60.3 | 64.4 | 28.5 |
| MFCL | NeurIPS 2023 | 32.9 | 44.1 | 25.9 | 31.4 | 43.2 | 26.4 | 30.7 | 42.8 | 28.8 |
| MFCL-△ | NeurIPS 2023 | 52.8 | 61.3 | 31.9 | 52.1 | 60.4 | 32.8 | 51.9 | 59.1 | 33.2 |
| FedCBC | MM 2024 | 17.6 | 22.9 | 34.7 | 17.4 | 23.1 | 35.3 | 15.9 | 21.0 | 36.6 |
| LANDER | CVPR 2024 | 34.8 | 48.5 | 27.6 | 33.4 | 45.2 | 28.5 | 32.1 | 43.5 | 30.5 |
| FedMGP | KDD 2024 | 23.4 | 33.1 | 34.9 | 21.8 | 32.7 | 35.6 | 20.9 | 32.4 | 36.9 |
| PiLoRA | ECCV 2024 | 69.6 | 72.1 | 26.7 | 63.7 | 65.4 | 27.8 | 63.0 | 68.0 | 29.4 |
| PiLoRA-△ | ECCV 2024 | 68.7 | 70.3 | 26.6 | 63.8 | 66.2 | 28.0 | 63.0 | 67.6 | 29.8 |
| MultiFCL | This Paper | **70.7** | **76.1** | **16.4** | **68.9** | **74.6** | **18.5** | **67.9** | **72.8** | **17.5** |

Table 2: Main results of performance comparison on CIFAR100 with 10 tasks in quality-based partitioning, where -△ means using CLIP as base model. The best results are shown in bold.

| | Methods | $\alpha = 6$ | | | $\alpha = 4$ | | | $\alpha = 2$ | | |
|---|---|---|---|---|---|---|---|---|---|---|
| | | Last | Avg. | $\mathcal{F}$ | Last | Avg. | $\mathcal{F}$ | Last | Avg. | $\mathcal{F}$ |
| ViT | ICLR 2021 | 64.6 | 67.8 | 20.9 | 63.5 | 66.4 | 22.0 | 63.0 | 66.5 | 21.6 |
| CLIP | ICML 2021 | 64.3 | 67.7 | 21.4 | 63.6 | 66.7 | 21.6 | 62.7 | 64.4 | 22.5 |
| MFCL | NeurIPS 2023 | 30.4 | 39.8 | 33.6 | 24.6 | 33.1 | 34.5 | 15.9 | 21.2 | 35.9 |
| MFCL-△ | NeurIPS 2023 | 49.1 | 57.9 | 29.4 | 48.7 | 56.2 | 29.9 | 47.1 | 55.3 | 31.2 |
| FedCBC | MM 2024 | 17.5 | 21.8 | 27.4 | 15.5 | 20.6 | 27.9 | 15.1 | 19.5 | 28.5 |
| LANDER | CVPR 2024 | 20.3 | 26.7 | 28.9 | 19.8 | 24.8 | 28.6 | 18.7 | 24.4 | 29.3 |
| FedMGP | KDD 2024 | 19.8 | 23.5 | 30.3 | 13.6 | 17.3 | 31.6 | 12.6 | 16.5 | 32.5 |
| PiLoRA | ECCV 2024 | 70.5 | 74.8 | 22.8 | 63.8 | 68.2 | 24.4 | 63.6 | 69.5 | 25.7 |
| PiLoRA-△ | ECCV 2024 | 69.8 | 73.2 | 23.4 | 64.6 | 69.2 | 24.6 | 64.5 | 70.1 | 25.3 |
| MultiFCL | This Paper | **71.0** | **76.1** | **15.9** | **70.3** | **75.3** | **15.9** | **69.7** | **75.2** | **17.1** |

consistency with our base model for comparison. Specifically, we set up 10 clients, with each client owning 4 experts and performing 5 epochs of local training. For non-PTM methods, we use a learning rate of 1e-2 and conduct 100 communication rounds per task. For PTM methods, we use a learning rate of 1e-5 and perform 5 communication rounds per task.

## 4.2 Main Results

We compare MultiFCL with other baselines, the results are shown in Table 1 and Table 2. For FCL methods trained from scratch, MFCL and LANDER use generators deployed on the server to synthesize data and apply knowledge distillation to transfer knowledge to the global model, yet without guidance from real samples the generators struggle to produce high-quality data. FedCBC builds a binary classifier for each class and performs selective knowledge fusion between local and global models in an attempt to balance new and old knowledge, yet these classifiers focus only on single-class decision boundaries and cannot capture the complex semantic relationships among multiple classes. In addition, their accuracy is much lower than that of pre-trained ViT and CLIP, which demonstrates the powerful generalization ability of PTMs.

For PTM-based FCL methods, FedMGP has lower accuracy under quality-based data partitioning due to difficulties in finding task similarities. Its poor performance stems from the limitations of its prompt pool mechanism in FCL, especially when tasks differ greatly. Furthermore, its data split differs substantially from ours because the first task contains more classes giving the model an opportunity to overfit early categories, so it performs poorly in our experiments. PiLoRA learns independent low-rank LoRA parameters for each task and adds regularization to the loss to mitigate

forgetting, but it lacks a multi-scale cross-task knowledge fusion and dynamic integration mechanism, its forgetting rate reaches 29.4% under distribution heterogeneity ($\beta = 0.05$) and its accuracy drops to only 63.6% under quality heterogeneity ($\alpha = 2$).

Meanwhile, we also replace the base model in MFCL and PiLoRA with CLIP to create MFCL-CLIP and PiLoRA-CLIP. CLIP's strong alignment of vision languages improves the accuracy of MFCL,it still relies on the generator to synthesize old samples of limited quality.And PiLoRA-CLIP still lacks cross-task knowledge integration. In contrast, direct fine-tuning of ViT and CLIP does not include any anti-forgetting design, whereas MultiFCL achieve sustained and robust performance gains across all settings and experimental results confirm its effectiveness and expressiveness.

### 4.3 Ablation Study

We perform the ablation study of MultiFCL on CIFAR100 with 10 Tasks, the results are shown in Table 3. The remaining experiments can be found in the appendix. Specifically, when no modules are added, the results correspond to the PTM itself, which establishes a clear baseline for comparison. When multiple adapters are added for fine-tuning on the first task, the average accuracy increases, demonstrating that task-specific adaptation allows the PTM to quickly adapt to downstream image classification tasks and exhibit its strong generalization ability.

When multi-modal features are added to initialize prototypes, MultiFCL supplements old class knowledge, thereby mitigating catastrophic forgetting in the current task. This addition yields a further gain in accuracy, confirming the benefit of richer prototype representations. Furthermore, when multi-experts are incorporated for training using $\mathcal{L}_{\text{experts}}$, each expert specializes in a different scale of features and together provide multi-scale knowledge, helping the model better distinguish differences between samples, resulting in higher accuracy. After incorporating multi-teacher self-distillation $\mathcal{L}_{\text{self-kd}}$, MultiFCL transfers knowledge from all experts to the final expert, reducing computational overhead during testing while consolidating expert knowledge, ultimately improving classification accuracy. The cumulative results of these experiments validate the design of MultiFCL and illustrate how each component contributes to stable and effective continual learning under federated and heterogeneous conditions.

Table 3: Ablation results of performance comparison on CIFAR100 with 10 tasks. The best results are shown in bold.

| Adapter | Init. | $\mathcal{L}_{\text{experts}}$ | $\mathcal{L}_{\text{self-kd}}$ | $\alpha = 6$ | $\alpha = 4$ | $\alpha = 2$ | $\beta = 0.5$ | $\beta = 0.1$ | $\beta = 0.05$ |
|---------|-------|------------|------------|----------|----------|----------|-----------|-----------|------------|
|         |       |            |            | 64.3 | 63.5 | 63.0 | 65.0 | 63.3 | 60.3 |
| ✓       |       |            |            | 67.6 | 64.9 | 65.2 | 66.0 | 64.5 | 63.1 |
| ✓       | ✓     |            |            | 68.5 | 67.0 | 66.9 | 67.1 | 65.3 | 64.7 |
| ✓       | ✓     | ✓          |            | 70.1 | 68.7 | 68.6 | 68.9 | 67.8 | 66.3 |
| ✓       | ✓     | ✓          | ✓          | **71.0** | **70.3** | **69.7** | **70.0** | **68.9** | **67.9** |

### 4.4 Adapter Frozen Study

Figure 4 shows the experimental results for freezing adapters versus keeping them trainable on CIFAR-100 with $\beta = 0.05$ and $\alpha = 2$. When adapters continue to be trained in later tasks, the model's performance drops rapidly because prolonged training leads to catastrophic forgetting, which undermines generalization and compromises plasticity. Furthermore, due to client data heterogeneity, the adapters tend to drift during training, resulting in a severe loss of plasticity. Under these conditions of extreme heterogeneity and limited class samples, the model experiences pronounced forgetting as adapter updates quickly diverge from the task's optimal direction, causing the PTM's strong generalization ability to be lost. Therefore, freezing the adapters after the first task not only reduces computational cost and communication overhead but also effectively preserves the model's generalization capacity, facilitating the transfer of previous knowledge to new tasks.

### 4.5 Expert Study

We use a uniform hierarchical strategy to divide the 12 Transformer layers into 4 modules, each containing 3 layers that correspond to shallow texture features, mid-level semantic features, and

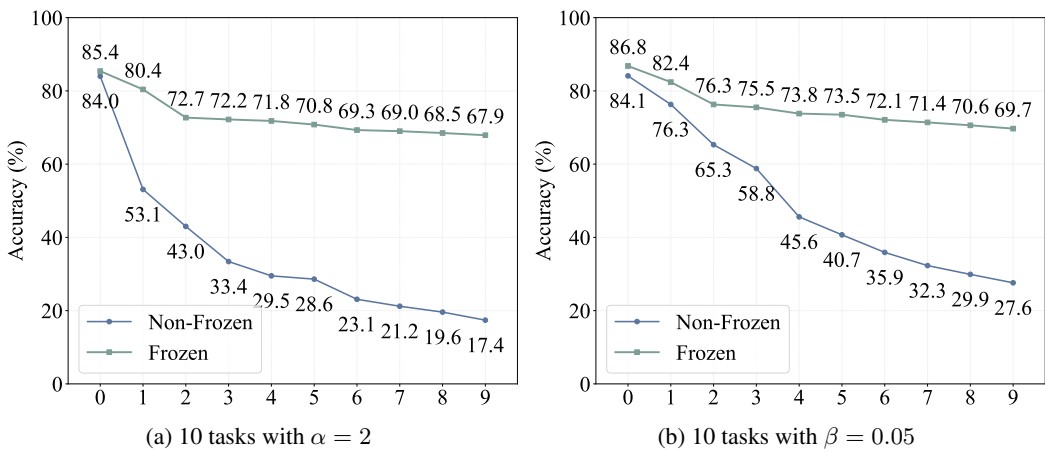

Figure 4: Adapter frozen results in the last task on CIFAR100, the x-axis represents which task it is.

Table 4: Expert number results of performance comparison on ImageNet-R with 10 Tasks. The best results are shown in bold.

| | $\beta = 0.5$ | | | $\beta = 0.1$ | | | $\beta = 0.05$ | | |
|---|---|---|---|---|---|---|---|---|---|
| | Last | Avg. | $\mathcal{F}$ | Last | Avg. | $\mathcal{F}$ | Last | Avg. | $\mathcal{F}$ |
| 12 Experts | 72.5 | 78.3 | 14.7 | 72.5 | 78.2 | 15.3 | 72.2 | 78.1 | 15.8 |
| 6 Experts | 73.2 | 78.7 | 14.3 | 73.2 | 78.7 | 14.6 | 72.7 | 78.3 | 15.6 |
| 4 Experts | **74.0** | **79.5** | **13.9** | **73.5** | **79.2** | **14.3** | **72.9** | **79.1** | **15.4** |
| 2 Experts | 72.6 | 78.4 | 14.9 | 72.3 | 78.4 | 15.5 | 72.2 | 78.1 | 16.1 |

deep abstract features, and we insert an expert into each module. It is important to note that this expert allocation strategy is not unique. We investigate how the number of experts affects model performance on distribution-based splits of ImageNet-R, and the results are shown in Table 4. With 12 experts, there is one per layer, with 6 experts, there is one every 1 layer. And with 2 experts, there is one every 5 layers. The results show that having too many experts prevents the final expert from fully learning the relevant knowledge, while having too few experts limits the amount of knowledge that can be transferred. Therefore, it is essential to choose an appropriate number of experts to ensure that sufficient and important knowledge is passed on to the final expert.

## 5 Conclusion

In this paper, we introduce MultiFCL, a novel FCL framework to address the stability-plasticity trade-off in FCL while leveraging the strengths of PTMs. Specifically, to maintain the stability of PTMs, MultiFCL incorporates lightweight adapters for efficient task adaption, and prevents catastrophic forgetting by freezing them during fine-tuning. To further preserve knowledge retention, it employs multi-modal prototype initialization, incorporating semantic features and sample features to enhance class representations. On the other hand, to strengthen the plasticity of PTMs, MultiFCL employs a multi-expert training mechanism, which combines multi-scale feature learning with multi-teacher self-distillation to improve knowledge transfer and fusion. Experimental results demonstrate that our proposed MultiFCL achieves state-of-the-art performance across multiple datasets and configurations, highlighting its effectiveness in FCL.

## Acknowledgment

This work is supported by the National Key Research and Development Program (No. 2022YFB3103202) of China.

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

# A Supplemental Experiments

## A.1 Main Results Supplement

Table 5 and Table 6 presents the results on CIFAR100 with 5 tasks. It can be observed that MultiFCL consistently achieves the best performance across all data partitioning settings, demonstrating its superior ability to adapt to new tasks while maintaining strong generalization.

Specifically, as $\alpha$ decreases, MultiFCL exhibits the smallest performance drop, indicating that it effectively leverages PTM pre-training characteristics along with multi-scale experts and self-distillation to mitigate catastrophic forgetting and enhance model stability.

Moreover, as $\beta$ decreases, the data distribution becomes increasingly heterogeneous, leading to an overall decline in model accuracy. However, MultiFCL still demonstrates the highest robustness. For instance, under the most extreme partitioning condition ($\beta = 0.05$), MultiFCL achieves 68.1% (Last) and 72.4% (Avg.), significantly outperforming PiLoRA (63.5% and 68.3%) and FedViT (60.3% and 64.4%). This result highlights the strong learning capability of MultiFCL in low-data and highly heterogeneous federated environments, effectively preventing severe knowledge forgetting during the FCL process.

Table 5: Main results of performance comparison on CIFAR100 with 5 tasks in distribution-based partitioning, where -△ means using CLIP as base model. The best results are shown in bold.

|  | Methods | $\beta = 0.5$ | | | $\beta = 0.1$ | | | $\beta = 0.05$ | | |
|---|---|---|---|---|---|---|---|---|---|---|
|  |  | Last | Avg. | $\mathcal{F}$ | Last | Avg. | $\mathcal{F}$ | Last | Avg. | $\mathcal{F}$ |
| ViT | ICLR 2021 | 65.0 | 68.2 | 25.1 | 63.3 | 66.7 | 25.7 | 60.3 | 64.4 | 26.0 |
| CLIP | ICML 2021 | 64.5 | 67.6 | 25.8 | 63.6 | 67.3 | 25.9 | 60.5 | 64.1 | 26.3 |
| MFCL | NeurIPS 2023 | 36.9 | 47.2 | 25.7 | 32.5 | 44.8 | 26.4 | 30.7 | 41.9 | 27.2 |
| FedCBC | MM 2024 | 23.9 | 23.0 | 30.5 | 20.4 | 24.2 | 31.1 | 17.8 | 22.9 | 32.5 |
| LANDER | CVPR 2024 | 25.8 | 26.6 | 28.7 | 24.6 | 26.3 | 29.4 | 24.7 | 26.0 | 30.7 |
| FedMGP | KDD 2024 | 24.4 | 35.2 | 31.2 | 24.4 | 33.5 | 32.1 | 24.7 | 31.4 | 32.6 |
| PiLoRA | ECCV 2024 | 64.2 | 68.6 | 26.9 | 66.8 | 69.5 | 27.3 | 63.5 | 68.3 | 27.4 |
| PiLoRA-△ | ECCV 2024 | 66.3 | 68.2 | 26.8 | 66.4 | 68.8 | 27.1 | 63.4 | 68.0 | 27.6 |
| MultiFCL | This Paper | **72.4** | **75.5** | **12.6** | **70.0** | **73.1** | **12.7** | **68.1** | **72.4** | **12.6** |

Table 6: Main results of performance comparison on CIFAR100 with 5 tasks in quality-based partitioning, where -△ means using CLIP as base model. The best results are shown in bold.

|  | Methods | $\alpha = 6$ | | | $\alpha = 4$ | | | $\alpha = 2$ | | |
|---|---|---|---|---|---|---|---|---|---|---|
|  |  | Last | Avg. | $\mathcal{F}$ | Last | Avg. | $\mathcal{F}$ | Last | Avg. | $\mathcal{F}$ |
| ViT | ICLR 2021 | 63.4 | 66.9 | 27.4 | 62.8 | 66.4 | 28.1 | 62.3 | 65.3 | 28.7 |
| CLIP | ICML 2021 | 63.4 | 67.2 | 27.7 | 63.1 | 66.7 | 28.0 | 62.6 | 65.8 | 28.5 |
| MFCL | NeurIPS 2023 | 35.4 | 43.7 | 30.6 | 31.3 | 35.8 | 31.4 | 21.9 | 26.4 | 32.8 |
| FedCBC | MM 2024 | 20.3 | 25.7 | 28.9 | 18.5 | 22.4 | 29.5 | 17.1 | 22.5 | 31.0 |
| LANDER | CVPR 2024 | 36.8 | 45.5 | 27.8 | 32.6 | 37.8 | 28.0 | 27.6 | 31.2 | 29.1 |
| FedMGP | KDD 2024 | 19.8 | 24.3 | 30.6 | 16.6 | 19.1 | 31.2 | 14.7 | 18.9 | 32.0 |
| PiLoRA | ECCV 2024 | 66.4 | 72.9 | 28.4 | 64.4 | 68.9 | 29.1 | 64.2 | 69.3 | 29.6 |
| PiLoRA-△ | ECCV 2024 | 65.5 | 72.1 | 28.9 | 64.1 | 68.8 | 29.3 | 63.9 | 68.7 | 29.8 |
| MultiFCL | This Paper | **69.2** | **74.1** | **10.4** | **69.9** | **74.6** | **11.0** | **68.5** | **73.5** | **12.3** |

## A.2 Ablation Results Supplement

To further analyze the contributions of key components in MultiFCL, we conduct ablation studies on CIFAR100. Table 7 illustrate the impact of each module on the final model performance, including adapter, prototype initialization, $\mathcal{L}_{\text{experts}}$, and $\mathcal{L}_{\text{self-kd}}$.

Table 7: Ablation results of performance comparison on CIFAR100 with 5 Tasks. The best results are shown in bold.

| Adapter | Init. | $\mathcal{L}_{\text{experts}}$ | $\mathcal{L}_{\text{self-kd}}$ | $\alpha = 6$ | $\alpha = 4$ | $\alpha = 2$ | $\beta = 0.5$ | $\beta = 0.1$ | $\beta = 0.05$ |
|---|---|---|---|---|---|---|---|---|---|
| | | | | 63.4 | 62.8 | 62.3 | 65.5 | 63.8 | 61.2 |
| ✓ | | | | 65.7 | 64.2 | 62.8 | 67.4 | 65.0 | 63.2 |
| ✓ | ✓ | | | 66.5 | 65.7 | 63.8 | 68.2 | 66.9 | 64.6 |
| ✓ | ✓ | ✓ | | 68.1 | 68.6 | 67.3 | 70.9 | 68.7 | 66.7 |
| ✓ | ✓ | ✓ | ✓ | **69.2** | **69.9** | **68.5** | **72.4** | **70.0** | **68.1** |

Table 8: Few-shot results of performance comparison on CUB-200-2011 with 10 tasks in distribution-based partitioning, where -$\triangle$ means using CLIP as base model. The best results are shown in bold.

| | Methods | $\beta = 0.5$ | | | $\beta = 0.1$ | | | $\beta = 0.05$ | | |
|---|---|---|---|---|---|---|---|---|---|---|
| | | Last | Avg. | $\mathcal{F}$ | Last | Avg. | $\mathcal{F}$ | Last | Avg. | $\mathcal{F}$ |
| ViT | ICLR 2021 | 73.5 | 75.6 | 29.7 | 71.2 | 74.9 | 30.3 | 70.9 | 73.2 | 31.7 |
| CLIP | ICML 2021 | 73.0 | 74.9 | 30.0 | 71.4 | 75.0 | 30.8 | 70.9 | 73.3 | 32.0 |
| MFCL | NeurIPS 2023 | 22.8 | 27.6 | 33.6 | 21.1 | 25.7 | 34.1 | 20.6 | 25.9 | 35.3 |
| FedCBC | MM 2024 | 20.2 | 29.5 | 30.5 | 19.9 | 27.4 | 31.7 | 17.1 | 26.3 | 32.8 |
| PiLoRA | ECCV 2024 | 63.0 | 73.5 | 29.6 | 53.9 | 68.0 | 30.7 | 57.3 | 69.3 | 31.5 |
| PiLoRA-$\triangle$ | ECCV 2024 | 62.6 | 72.2 | 30.2 | 53.6 | 67.3 | 31.6 | 57.5 | 69.4 | 31.6 |
| MultiFCL | This Paper | **78.5** | **84.5** | **16.7** | **77.7** | **84.9** | **18.2** | **77.2** | **84.9** | **18.0** |

Introducing adapters improves performance across all settings, with 5 tasks improving by 2.3%. And with the addition of prototype initialization, it shows further accuracy gains. After incorporating $\mathcal{L}_{\text{experts}}$, accuracy for 5 tasks improves by 1.6% - 3.5%, confirming that it facilitates knowledge sharing across tasks and enhances the model's ability to handle imbalanced data distributions.

Finally, with $\mathcal{L}_{\text{self-kd}}$, the model achieves the highest accuracy. Overall, these results validate that MultiFCL, by integrating multiple critical modules, significantly boosts model performance in FCL scenarios across various data partitioning settings.

## A.3 Few-Shot Study

In FL, resource-constrained clients may have only a limited number of samples for training, and this data scarcity exacerbates the challenge of adapting to new tasks over the long-term learning process. To address this, we conduct the few-shot study, exploring the performance of existing methods under few-shot conditions. Specifically, we experiment on the few-shot dataset CUB-200-2011. Under the distribution-based data partitioning, each client has a few dozen samples, while under the quality-based partitioning, each client only has three samples per class. The results are shown in Table 8 and Table 9.

It can be observed that MultiFCL demonstrates the most significant advantage in the few-shot setting. Since PTMs are pre-trained on large-scale datasets, they can achieve expressive results with only a small number of samples. In contrast, FCL methods that train models from scratch tend to over-fit in the few-shot setting, leading to a significant performance decline. Meanwhile, MultiFCL enhances the model's ability to adapt to new tasks through multi-teacher self-distillation and multi-scale feature learning, preventing excessive reliance on the limited sample information. Experimental results show that MultiFCL effectively alleviates adaptation challenges in few-shot settings, maintaining its performance even in data-scarce scenarios.

What's more, Table 10 and Table 11 presents the results on CUB-200-2011 with 5 tasks. MultiFCL consistently achieves the best performance across all data partitioning settings. For instance, when $\alpha = 6$, MultiFCL attains 66.6% (Last) and 75.9% (Avg.) accuracy, outperforming PiLoRA by 27.5% and 24.8%, respectively, and surpassing FedViT by 5.6% and 6.8%. Even under extremely low-data conditions, MultiFCL effectively leverages pre-trained knowledge while enhancing model

Table 9: Few-shot results of performance comparison on CUB-200-2011 with 10 tasks in quality-based partitioning, where -△ means using CLIP as base model. The best results are shown in bold.

| | Methods | $\alpha = 6$ | | | $\alpha = 4$ | | | $\alpha = 2$ | | |
| --- | --- | --- | --- | --- | --- | --- | --- | --- | --- | --- |
| | | Last | Avg. | $\mathcal{F}$ | Last | Avg. | $\mathcal{F}$ | Last | Avg. | $\mathcal{F}$ |
| ViT | ICLR 2021 | 61.0 | 69.1 | 33.9 | 57.9 | 64.3 | 34.5 | 53.1 | 56.5 | 35.2 |
| CLIP | ICML 2021 | 61.1 | 69.3 | 34.0 | 57.6 | 64.5 | 34.4 | 53.2 | 56.6 | 35.5 |
| MFCL | NeurIPS 2023 | 18.8 | 21.5 | 34.4 | 13.6 | 19.9 | 35.0 | 10.8 | 17.3 | 35.7 |
| FedCBC | MM 2024 | 17.4 | 21.4 | 34.3 | 15.9 | 21.0 | 35.2 | 15.2 | 20.5 | 35.6 |
| PiLoRA | ECCV 2024 | 39.1 | 51.1 | 35.7 | 43.9 | 57.0 | 36.4 | 53.2 | 65.9 | 37.2 |
| PiLoRA-△ | ECCV 2024 | 40.3 | 51.6 | 36.1 | 44.0 | 57.2 | 37.3 | 53.2 | 65.8 | 37.0 |
| MultiFCL | This Paper | **66.6** | **75.9** | **24.0** | **64.4** | **73.5** | **25.2** | **57.0** | **69.0** | **26.0** |

Table 10: Few-shot results of performance comparison on CUB-200-2011 with 5 tasks in distribution-based partitioning, where -△ means using CLIP as base model. The best results are shown in bold.

| | Methods | $\beta = 0.5$ | | | $\beta = 0.1$ | | | $\beta = 0.05$ | | |
| --- | --- | --- | --- | --- | --- | --- | --- | --- | --- | --- |
| | | Last | Avg. | $\mathcal{F}$ | Last | Avg. | $\mathcal{F}$ | Last | Avg. | $\mathcal{F}$ |
| ViT | ICLR 2021 | 73.2 | 75.0 | 29.9 | 72.0 | 74.9 | 30.5 | 70.4 | 73.6 | 31.1 |
| CLIP | ICML 2021 | 73.0 | 75.2 | 30.4 | 72.1 | 75.0 | 30.7 | 71.3 | 73.9 | 32.0 |
| MFCL | NeurIPS 2023 | 23.9 | 28.6 | 30.3 | 21.2 | 26.7 | 32.6 | 20.5 | 26.0 | 32.1 |
| FedCBC | MM 2024 | 21.5 | 30.7 | 30.2 | 19.5 | 27.3 | 30.8 | 18.3 | 27.0 | 31.9 |
| PiLoRA | ECCV 2024 | 61.0 | 70.8 | 27.6 | 55.6 | 66.0 | 28.3 | 58.4 | 68.3 | 29.0 |
| PiLoRA-△ | ECCV 2024 | 60.7 | 70.2 | 28.0 | 55.7 | 66.4 | 28.3 | 58.3 | 68.0 | 29.3 |
| MultiFCL | This Paper | **77.4** | **83.7** | **15.0** | **77.8** | **83.8** | **14.8** | **77.8** | **84.1** | **15.6** |

generalization through multi-scale learning and self-distillation mechanisms. As a result, it maintains stable and competitive performance even under highly heterogeneous data distributions.

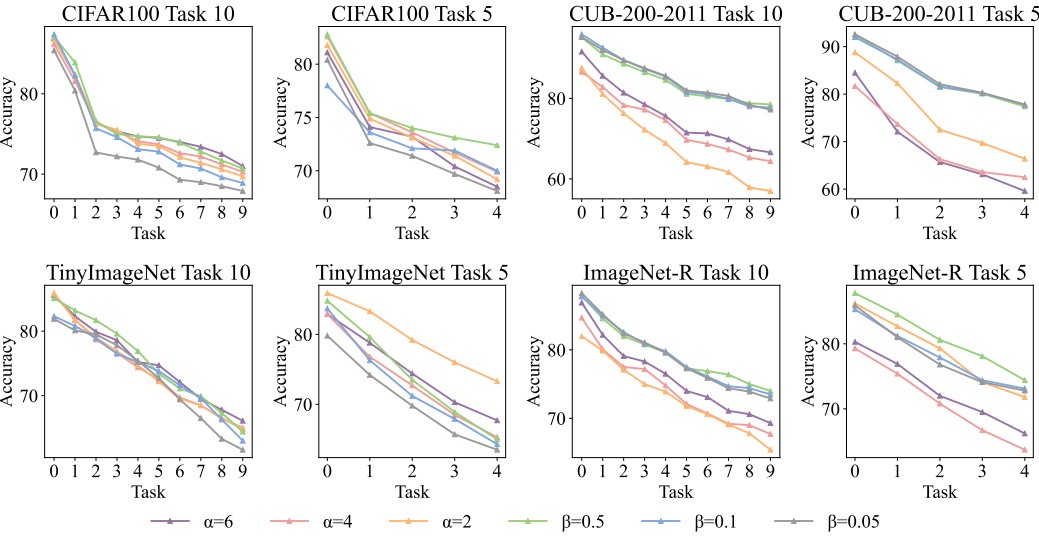

Figure 5: The performance of MultiFCL per task in the 5 and 10 tasks on all datasets.

## A.4 Incremental Trend Study

Figure 5 presents the FCIL results of MultiFCL across all datasets and data splits, where fine-tuning the adapters on the first task enables the PTM to achieve highly expressive performance. As

Table 11: Few-shot results of performance comparison on CUB-200-2011 with 5 tasks in quality-based partitioning, where -△ means using CLIP as base model. The best results are shown in bold.

| | Methods | $\alpha = 6$ | | | $\alpha = 4$ | | | $\alpha = 2$ | | |
| --- | --- | --- | --- | --- | --- | --- | --- | --- | --- | --- |
| | | Last | Avg. | $\mathcal{F}$ | Last | Avg. | $\mathcal{F}$ | Last | Avg. | $\mathcal{F}$ |
| ViT | ICLR 2021 | 60.3 | 68.9 | 30.2 | 58.4 | 65.5 | 31.0 | 54.6 | 58.7 | 31.7 |
| CLIP | ICML 2021 | 60.4 | 68.2 | 30.4 | 58.6 | 65.7 | 31.1 | 55.0 | 59.2 | 32.0 |
| MFCL | NeurIPS 2023 | 20.4 | 22.5 | 33.3 | 18.7 | 20.3 | 33.8 | 15.2 | 18.6 | 35.1 |
| FedCBC | MM 2024 | 19.5 | 21.6 | 32.3 | 16.9 | 22.1 | 33.9 | 15.2 | 21.2 | 34.5 |
| PiLoRA | ECCV 2024 | 49.8 | 59.3 | 30.9 | 50.9 | 61.9 | 31.6 | 62.5 | 72.9 | 32.7 |
| PiLoRA-△ | ECCV 2024 | 50.0 | 59.9 | 30.1 | 51.0 | 66.3 | 31.7 | 62.4 | 67.6 | 32.4 |
| MultiFCL | This Paper | **66.4** | **75.9** | **19.2** | **62.5** | **69.6** | **22.4** | **59.6** | **69.0** | **24.9** |

the number of tasks increases and the model must recognize all previously seen classes, accuracy gradually declines. However, from the second through fourth tasks, by using multiple experts to help the model quickly understand the knowledge of new and old tasks, PTM is able to distinguish old classes from new ones, maintaining a balance between stability and plasticity.

## A.5 Extreme Heterogeneity Study

In our experiments, distribution-based partitioning corresponds to Non-IID splitting, and the value of $\beta$ determines the degree of heterogeneity, with smaller values producing more heterogeneous data. We evaluate MultiFCL's performance on the final ImageNet-R task under extreme client distribution heterogeneity, as shown in Figure 6. The results show that MultiFCL maintains expressive performance even at $\beta = 0.05$. In addition, we test the model at $\beta$ values of 0.01, 0.005, and 0.001 and found that MultiFCL handles all of these scenarios effectively. Even in the most extreme cases the model's accuracy fluctuates by only 1.4%. This robustness comes from clients learning diverse perspectives through experts and sharing knowledge internally, while FL enables external knowledge exchange from multiple angles, allowing the model to quickly adapt to new tasks and remain highly resilient.

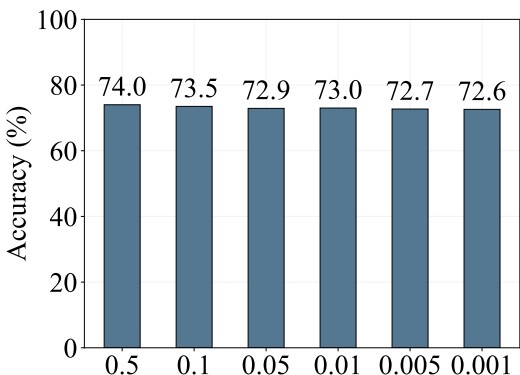

Figure 6: Extreme heterogeneity results on ImageNet-R with 10 tasks, the x-axis represents the value of $\beta$.

## A.6 Loss Weight Study

We explore the relationship between weights of two loss functions during client local training, with the CIFAR100 results shown in Table 12, and the CUB-200-2011 results are shown in Table 13. Specifically, we conduct experiments by selecting values for $\lambda'_{\text{expert}}$ from {0.1, 0.3, 0.5, 0.7, 0.9}. As the value decreases, the importance of multi-teacher self-distillation $\mathcal{L}_{\text{self-kd}}$ becomes smaller. From the results, it is evident that the performance of MultiFCL on CIFAR100 experiences only minor fluctuations, and different loss function weights do not significantly impact the overall performance of the model. This indicates that the coordination between multi-teacher self-distillation and multi-scale feature learning losses is quite robust during training, allowing for flexible adjustments within a certain range without affecting the model's performance.

Additionally, when the value of $\lambda'_{\text{expert}}$ is around 0.5, MultiFCL achieves optimal performance. This suggests that learning knowledge from different experts helps the final expert better distinguish different samples, but the knowledge learned from former experts should not be too important, as this could cause the final expert to neglect the features it has extracted. By appropriately adjusting the

Table 12: Loss weight results of performance comparison on CIFAR100 with 10 tasks. The best results are shown in bold.

| | 10 Tasks | | | | | 5 Tasks | | | | |
|---|---|---|---|---|---|---|---|---|---|---|
| | 0.1 | 0.3 | 0.5 | 0.7 | 0.9 | 0.1 | 0.3 | 0.5 | 0.7 | 0.9 |
| $\alpha = 2$ | 68.6 | 69.2 | **69.7** | 69.6 | 69.5 | 67.9 | **68.2** | 68.1 | 68.0 | 67.7 |
| $\alpha = 4$ | 69.7 | 70.0 | **70.3** | 70.3 | 70.2 | 68.8 | 69.6 | **69.9** | 69.5 | 69.1 |
| $\alpha = 6$ | 69.0 | 69.3 | **71.0** | 69.6 | 68.5 | 68.3 | 68.9 | **69.2** | 68.9 | 68.7 |
| $\beta = 0.5$ | 69.1 | **70.7** | 70.6 | 69.9 | 70.1 | 71.9 | **72.4** | 72.0 | 71.8 | 71.5 |
| $\beta = 0.1$ | 68.3 | 68.7 | **68.9** | 68.8 | 68.7 | 69.3 | 69.7 | **70.0** | 69.9 | 69.5 |
| $\beta = 0.05$ | 66.5 | 66.9 | 67.2 | **67.9** | 67.8 | 67.7 | 68.0 | **68.1** | **68.1** | **68.1** |

Table 13: Loss weight results of performance comparison on CUB-200-2011 with 10 tasks. The best results are shown in bold.

| | 10 Tasks | | | | | 5 Tasks | | | | |
|---|---|---|---|---|---|---|---|---|---|---|
| | 0.1 | 0.3 | 0.5 | 0.7 | 0.9 | 0.1 | 0.3 | 0.5 | 0.7 | 0.9 |
| $\alpha = 2$ | 56.4 | 56.8 | **57.0** | 56.7 | 56.5 | 69.0 | 59.1 | 58.4 | **59.6** | 59.4 |
| $\alpha = 4$ | 64.1 | 64.2 | **64.4** | 64.3 | 64.0 | 61.9 | 62.1 | 62.4 | **62.5** | 62.1 |
| $\alpha = 6$ | 66.1 | 66.3 | 66.4 | **66.6** | 66.4 | 65.8 | 66.0 | **66.4** | 66.3 | 66.4 |
| $\beta = 0.5$ | 78.0 | 78.3 | 78.3 | **78.5** | 78.1 | 77.3 | **77.4** | 77.0 | 76.9 | 76.6 |
| $\beta = 0.1$ | 77.3 | 77.6 | **77.7** | 77.6 | 77.4 | 77.3 | 77.6 | **77.8** | 77.6 | 77.5 |
| $\beta = 0.05$ | 76.5 | 76.9 | **77.2** | 77.0 | 76.8 | 77.4 | 77.5 | **77.8** | **77.8** | 77.7 |

value, MultiFCL can be flexibly configured across different tasks and datasets, further enhancing the adaptability of the model.

## A.7 Long Tasks Study

We evaluate MultiFCL's performance on long task sequences by extending ImageNet-R to 20, 30, 40, and 50 tasks using distribution-based splits with $\beta = 0.05$, and the results are shown in Figure 7. It can be seen that the model remains stable across long sequences, even though each task contributes less new knowledge. The client experts effectively retain previous knowledge while assimilating new information, allowing the model to enhance plasticity without sacrificing stability. As a result, MultiFCL exhibits only a 1.3% fluctuation in accuracy over these extended task sequences, demonstrating its effectiveness and robustness.

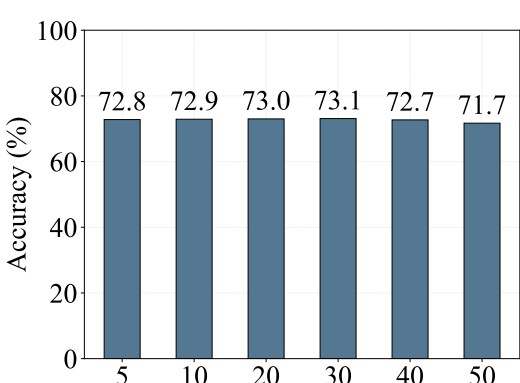

Figure 7: Long task results on ImageNet-R with $\beta = 0.05$, the x-axis is the number of tasks.

## A.8 TIL Study

We analyze the TIL results of MultiFCL on CIFAR-100 under both 10-task and 5-task settings, as shown in Tables 14 and 15. MultiFCL consistently achieves the best performance. The tables show that TIL tasks are easier than CIL tasks since they do not require evaluating the model's performance on previous tasks and focus only on the current task. MultiFCL is designed to enhance plasticity while preserving stability, and its performance on the final task far surpasses that of other PTM-based methods, demonstrating its superior plasticity and validating its effectiveness.

Table 14: TIL results of performance comparison on CIFAR100 with 10 tasks in distribution-based partitioning, where -△ means using CLIP as base model. The best results are shown in bold.

| | Methods | $\beta = 0.5$ | | | $\beta = 0.1$ | | | $\beta = 0.05$ | | |
|---|---|---|---|---|---|---|---|---|---|---|
| | | Last | Avg. | $\mathcal{F}$ | Last | Avg. | $\mathcal{F}$ | Last | Avg. | $\mathcal{F}$ |
| ViT | ICLR 2021 | 70.6 | 75.7 | 25.4 | 69.2 | 72.9 | 25.9 | 68.8 | 72.7 | 26.4 |
| CLIP | ICML 2021 | 70.7 | 76.0 | 25.9 | 69.8 | 72.4 | 26.1 | 68.2 | 72.4 | 25.9 |
| PiLoRA | ECCV 2024 | 74.4 | 75.4 | 24.3 | 67.3 | 72.9 | 24.1 | 74.8 | 72.8 | 25.7 |
| PiLoRA-△ | ECCV 2024 | 74 6 | 75.5 | 24.1 | 67.0 | 73.3 | 25.0 | 74.7 | 64.3 | 26.6 |
| MultiFCL | This Paper | **75.3** | **79.5** | **10.6** | **74.0** | **77.8** | **12.8** | **72.5** | **76.5** | **11.9** |

Table 15: TIL results of performance comparison on CIFAR-100 with 5 tasks in distribution-based partitioning, where -△ means using CLIP as base model. The best results are shown in bold.

| | Methods | $\beta = 0.5$ | | | $\beta = 0.1$ | | | $\beta = 0.05$ | | |
|---|---|---|---|---|---|---|---|---|---|---|
| | | Last | Avg. | $\mathcal{F}$ | Last | Avg. | $\mathcal{F}$ | Last | Avg. | $\mathcal{F}$ |
| ViT | ICLR 2021 | 71.2 | 74.7 | 15.5 | 70.8 | 72.6 | 15.0 | 70.0 | 71.8 | 16.8 |
| CLIP | ICML 2021 | 72.0 | 75.0 | 15.3 | 71.1 | 72.7 | 74.9 | 69.8 | 71.9 | 17.5 |
| PiLoRA | ECCV 2024 | 72.0 | 72.9 | 13.6 | 71.7 | 72.7 | 14.0 | 71.9 | 74.7 | 14.5 |
| PiLoRA-△ | ECCV 2024 | 72.6 | 73.0 | 13.0 | 71.7 | 72.7 | 14.2 | 72.1 | 74.3 | 13.9 |
| MultiFCL | This Paper | **77.8** | **78.8** | **1.0** | **75.5** | **74.9** | **1.5** | **76.0** | **76.9** | **2.4** |

## A.9  PTM Study

MultiFCL relies solely on CLIP's image-text alignment capability and can be extended to any PTM with such alignment. For example, we supplement Flava (Singh et al., 2022) and TinyCLIP ViT-39M/16 Text-19M (Wu et al., 2023) as our backbone model and report its last accuracy results in Table 16. MultiFCL maintains relatively stable performance across different PTMs. Beyond this, image-text alignment can also be achieved by adding projection layers, offering a pathway to extend compatibility with more PTMs. It can be seen that using TinyCLIP does not degrade MultiFCL's performance. Conversely, it remains highly stable, which demonstrates the potential for balancing resource constraints and performance requirements in resource-limited scenarios (e.g., edge devices).

Table 16: Comparison results of last accuracy on CIFAR100 with 10 tasks in both quality-based partitioning and distribution-based partitioning.

| Model | $\alpha = 6$ | $\alpha = 4$ | $\alpha = 2$ | $\beta = 0.5$ | $\beta = 0.1$ | $\beta = 0.05$ |
|---|---|---|---|---|---|---|
| CLIP-based | 71.0 | 70.3 | 69.7 | 70.7 | 68.9 | 67.9 |
| Flava-based | 70.4 | 69.9 | 69.7 | 69.5 | 68.2 | 67.5 |
| TinyCLIP ViT-39M-based | 70.8 | 70.2 | 69.5 | 70.9 | 70.0 | 69.2 |

## B  Data Partition Details

Referring to the data partitioning strategies described in PiLoRA (Guo et al., 2024), greater numbers of classes possessed by a client indicate higher data quality. Thus, in quality-based partitioning, each client is assigned data from $\alpha$ randomly selected classes. Specifically, the number of samples for each class on a client is determined by dividing the total number of samples for that class by the number of clients, ensuring that data across clients do not overlap. Therefore, a larger $\alpha$ corresponds to higher data quality for the clients. Additionally, In the distribution-based data partitioning, each client's data distribution is determined by the Dirichlet distribution, with smaller values of $\beta$ indicating higher data heterogeneity. Figure 8 shows the data distribution under different datasets and partitioning strategies in the last task, which the ID of clients and labels are only used to distinguish between different clients and labels. The code is available at `https://github.com/yang12318/MultiFCL`.

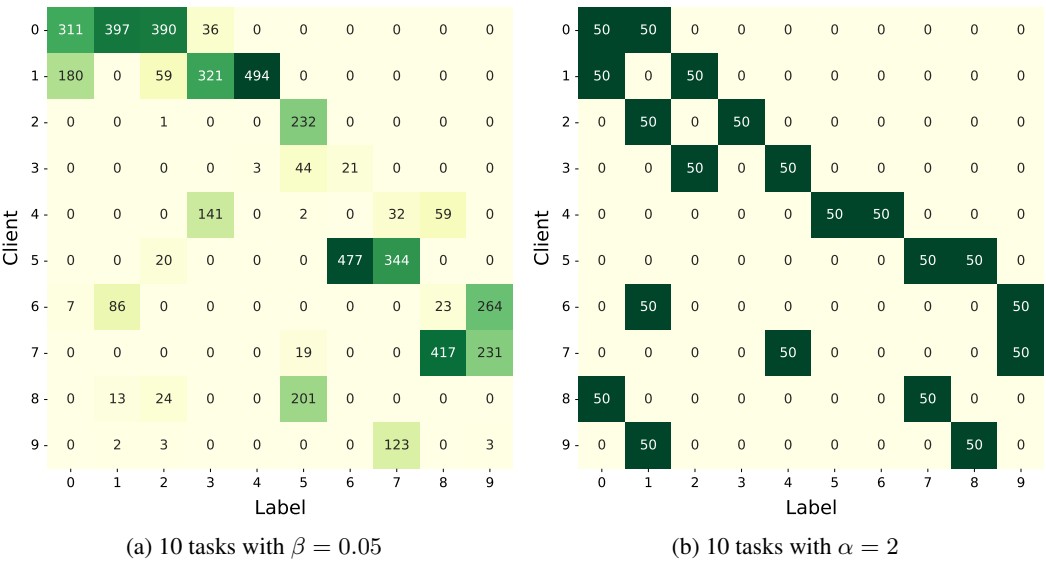

(a) 10 tasks with $\beta = 0.05$         (b) 10 tasks with $\alpha = 2$

Figure 8: The Data Partition in the Last Task on CIFAR100.

## C  Method Analysis

**Communication Cost.** The model parameters consist of the PTM, multiple experts, and adapters. Notably, the PTM does not require communication, and adapters only need to be exchanged during the first task. For instance, in a scenario with 10 tasks, the total number of expert parameters is 384,000, while the adapters account for 156,768 parameters. Approximately 0.54 million parameters are communicated during the first task, and only about 0.38 million in subsequent tasks. The following comparison of parameters in Table 17 with others demonstrates that MultiFCL is highly communication-efficient.

Table 17: Parameter Comparison.

| Model | Parameters |
|---|---|
| ResNet-18 | 11M |
| ResNet-50 | 23M |
| ResNet-101 | 43M |
| ViT-B/16-IN1K | 86M |
| ViT-B/16-CLIP | 86M |
| MultiFCL | 0.54/ 0.38M |

**Computational Efficiency.** In clients, for forward propagation, the multi-expert feature learning requires extracting features $\{h_m\}_{m=1}^{M}$ from $M$ modules. For a PTM with $L$ total layers divided into $M$ modules, the computational complexity remains: $\mathcal{C}_{\text{forward}} = O\left(\frac{L}{M} \cdot (d^2 \cdot S)\right) \times M = O(Ld^2S)$, where $S$ represents the sequence length, $d$ is the dimension of a prototype. For backward propagation, it is equivalent to forward propagation, which is $O(Ld^2S)$. In the server, for the first task, the server needs to aggregate experts and adapters uploaded by clients, with a computational complexity of $O(K \times M \times d) + O(K \times N \times D)$, where $D$ is the dimension of an adapter, and $N$ is the total number of adapters in a client. For the subsequent tasks, the server only needs to aggregate the prototype experts and adapters are frozen, with a computational complexity of $O(K \times M \times d)$.

We compare communication parameters and wall-clock per round for MultiFCL, PiLoRA, MFCL, and LANDER on CIFAR100. Each round includes 5 local epochs across 10 clients plus one server update, as shown in Table 18. In MFCL, the server must distribute three components to clients, including the current-task global model, previous-task global model, and generator. Although the global model uses ResNet-18 (11.2M), this still incurs substantial communication overhead. Meanwhile, despite its wall-clock per round remains low, it requires 100 communication rounds per task to achieve competitive results, whereas MultiFCL and PiLoRA attain expressive outcomes within merely 5 rounds. LANDER transmits the ResNet-18 as global model each round. Its integration of pre-trained CLIP and generator yields wall-clock per round comparable to PTM-based methods while requiring 100 rounds per task. PiLoRA communicates LoRA parameters and prototypes per round.

Although parameter volume stays low, continuous LoRA tuning prevents effective knowledge retention and increases computation time, resulting in suboptimal performance versus MultiFCL. MultiFCL freezes adapters after initial tasks, reducing communication parameters (0.54M to 0.38M) and wall-clock per round (228.29s to 150.24s) while achieving competitive performance within 5 communication rounds per task. Collectively, MultiFCL maintains computational efficiency while reducing communication costs versus existing methods.

Table 18: Comparison results of communication parameters and wall-clock per round on CIFAR100 with 10 tasks in distribution-based partitioning.

| Model | Parameters | Wall-clock |
|---|---|---|
| MFCL | 24.27M | 35.20s |
| LANDER | 11.20M | 169.65s |
| PiLoRA | 0.0584M | 207.10s |
| MultiFCL | 0.54/ 0.38M | 228.29/ 150.24s |

# D Algorithm

Algorithm 1 displays the detailed process in MultiFCL. It employs adapters to fine-tune the PTM and leverages the semantic features of old tasks to initialize new class prototypes. Then, it establishes multiple experts, employing feature learning loss and the multi-teacher dynamic self-distillation to transfer knowledge to the final expert.

---

**Algorithm 1** MultiFCL Framework

---

**Require:** Pre-trained model $f$, clients $K$, communication rounds $T$, tasks $\{\mathcal{T}_1, \mathcal{T}_2, \ldots, \mathcal{T}_\tau\}$, local epochs $E$

1: **for** each task $\mathcal{T}_t \in \{\mathcal{T}_1, \ldots, \mathcal{T}_\tau\}$ **do**
2:     **if** $t = 1$ **then**
3:         **for** round $r = 1$ **to** $T$ **do**
4:             **for** each client $k = 1$ **to** $K$ **in parallel do**
5:                 Fine-tune adapters on $D_k^1$
6:                 Upload $w_k^r$ and sample count $|X_k^1|$ to server
7:             **end for**
8:             **Server aggregates**: $w^r \leftarrow \sum_{k=1}^{K} \frac{|X_k^1|}{\sum |X_k^1|} w_k^r$
9:             **Server broadcasts** global adapter $w^r$ to clients
10:         **end for**
11:         Freeze global adapters $w^* \leftarrow w^T$
12:     **else**
13:         **for** each new class $c \in \mathcal{T}_t$ **do**
14:             Compute semantic similarity: $\text{sim} \leftarrow \frac{f'(x_1^c) \cdot f_{\text{text}}(C^{\mathcal{T}_{\text{old}}})}{\|f'(x_1^c)\| \|f_{\text{text}}(C^{\mathcal{T}_{\text{old}}})\|}$
15:             Initialize prototype: $\mathbf{p}_{\text{init}}^c \leftarrow \text{Attention}(f'(x_1^c), f_{\text{text}}(c_{\text{sim}}^{\mathcal{T}_{\text{old}}}), \mathbf{p}_{\text{old}}^c)$
16:         **end for**
17:     **end if**
18:     **for** round $r = 1$ **to** $T$ **do**
19:         **for** each client $k = 1$ **to** $K$ **in parallel do**
20:             **for** local epoch $e = 1$ **to** $E$ **do**
21:                 Extract multi-scale features: $\{h_m\}_{m=1}^M \leftarrow f'(x; w^*)$
22:                 Compute expert loss: $\mathcal{L} = \mathcal{L}_{\text{experts}} + \lambda'_{\text{expert}} \cdot \mathcal{L}_{\text{self-kd}}$
23:                 Update parameters: $\theta \leftarrow \theta - \eta \nabla_\theta(\mathcal{L})$
24:                 Update prototypes: $\mathbf{p}_{(j)}^c \leftarrow \frac{\mathbf{p}_{(j-1)}^c + f'(x_j^c)}{2}$
25:             **end for**
26:             Upload prototypes $\{\mathbf{p}_{k,m}^c\}_{m=1}^M$ and sample counts $|X_k^{c,t}|$ to server
27:         **end for**
28:         **Server aggregates prototypes**: $\mathbf{p}^{c,r} \leftarrow \sum_{k=1}^{K} \frac{|X_k^{c,t}|}{\sum |X_k^{c,t}|} \sum_m \mathbf{p}_{k,m}^{c,r}$
29:         **Server broadcasts** global prototypes $\mathbf{p}^r$ to clients
30:     **end for**
31: **end for**

---

# E    Limitations

Although the proposed MultiFCL framework delivers strong performance across a variety of heterogeneous data partitions and CL scenarios, it still has several limitations that call for further study. MultiFCL depends on large scale PTMs and their multi-scale feature modules, which may impose considerable computational and storage demands on devices with limited resources. The mechanisms for prototype initialization and semantic feature matching need refinement in extreme class imbalance or few sample scenarios to ensure that both new and existing classes are represented effectively. The current evaluation focuses mainly on image classification tasks and it remains necessary to investigate how MultiFCL can be applied to other modalities such as text and speech or extended to more complex tasks.

# F    Code of Ethics

This paper is guided by a commitment to user privacy, performs all model updates and feature extraction entirely in clients without collecting or storing any raw data. We provide a detailed description of each algorithmic component including multi-adapter fine-tuning, prototype initialization, multi-scale feature learning and multi-teacher adaptive self-distillation along with network architectures and hyperparameter settings to ensure that others can reproduce and validate our findings without accessing private data. During evaluation, we use publicly available datasets to emulate realistic distributions and prevent exposure of any individual-level information. All original data remains on local devices under the full control of each participant throughout the entire process.

# G    Societal Impacts

This paper presents work that aims to advance the field of deep learning. Our research explores how to leverage pre-trained models to enhance performance in FCL. By introducing mechanisms such as multi-adapter fine-tuning, multi-modal prototype initialization, multi-scale feature learning, and multi-teacher dynamic self-distillation, we improve the model's adaptability and generalization, ensuring performance is maintained while preventing catastrophic forgetting.

Since this work provides a novel perspective for deep learning from a technical standpoint, we recognize that with the broader application of such technologies, there may be some impacts to consider. For instance, FCL has the potential to play a critical role in fields like healthcare and the internet of things. This requires us to design methods that avoid bias or misleading decisions, and we are actively working towards that goal.

