# OpenReview forum: "Federated Continual Learning via Orchestrating Multi-Scale Expertise"
_NeurIPS.cc/2025/Conference — NeurIPS 2025 poster_

### Official Review · Reviewer_Bnjx · 2025-07-02

**Clarity:** 3
**Significance:** 3
**Originality:** 3
**Rating:** 4
**Confidence:** 4

**Summary:**

This paper proposes MultiFCL, a novel framework that leverages pre-trained models (PTMs) to address the challenges in Federated Continual Learning (FCL). To ensure stability, MultiFCL introduces lightweight adapters, which are frozen after the initial task training, to enhance PTM performance on downstream tasks. MultiFCL further utilizes prototypes as classifier weights, enabling classification by measuring the distance between class prototypes and sample features. Concurrently, to enhance plasticity, the framework integrates multi-scale feature learning with multi-teacher dynamic self-distillation. Extensive experiments are conducted to validate MultiFCL's state-of-the-art performance.

**Questions:**

as above

**Ethical Concerns:**

["NO or VERY MINOR ethics concerns only"]

**Final Justification:**

Thank you for your detailed and comprehensive rebuttal. After carefully reviewing your responses, most of my previous concerns have been addressed. However, I still have some concerns regarding the computational resources of this method, and therefore, I have decided to maintain my favorable rating.

**Limitations:**

yes

**Quality:**

3

**Strengths And Weaknesses:**

The paper is well-structured, logical, and innovative. However, there are still areas that need improvement and further clarification for the paper to be accepted.

Weakness:
1.	The paper's exploration of unique challenges inherent to FL is not enough. Furthermore, the integration and handling of various components primarily rely on basic weighted averaging, without delving into more FL-specific mechanisms. Consequently, the paper more focuses on Continual Learning, with less emphasis on novel contributions directly addressing the unique complexities of FCL.
2.	Figure 1 does not sufficiently illustrate the overall workflow of the proposed framework, which may lead to confusion. It is suggested that the authors further revise this figure to enhance its readability and facilitate a comprehensive understanding of the method's working process.
3.	While the framework comprises four components, the clarity of the authors' description for each part could be further improved. Specifically, a more detailed and structured explanation of their individual functions and interconnections within the overall workflow would significantly enhance reader comprehension.
4.	Regarding the Multi-Adapter Fine-Tuning: Firstly, this fine-tuning mechanism bears a strong resemblance to LoRA (Low-Rank Adaptation). It is recommended that the authors include relevant citations for LoRA and similar parameter-efficient fine-tuning techniques to provide appropriate context. Secondly, the rationale behind freezing these adapters immediately after the initial task training warrants further clarification. This design choice might potentially limit the method's knowledge transfer capability, especially when executing cross-domain tasks that could benefit from continuous adaptation of these modules. In summary, the overall motivation and implications behind the design of this specific component require more comprehensive elucidation, potentially including a discussion of trade-offs or supporting experimental justifications.
5.	Regarding the Multi-Modal Prototype Initialization component, my primary concern is: what is the source of the textual information associated with the images? Furthermore, a question arises regarding the fairness of using this multi-modal technique, which introduces additional knowledge for classification. Could the authors clarify if and how this extra knowledge source impacts the comparison with baselines that might not leverage such external information?
6.	Regarding the Multi-Scale Feature Learning component: Does every block need to perform a classification task? If so, is it required that each block calculates a prototype for its current layer for subsequent operations? These design choices raise concerns about their potential computational implications: would this significantly increase the client's local computational burden, thereby limiting the method's applicability in resource-constrained edge computing scenarios?
7.	Regarding the Multi-Teacher Dynamic Self-Distillation component: There is a concern that distilling all expert knowledge into a single expert might not necessarily translate into significant performance gains. Simultaneously, this component appears to introduce substantial computational overhead, which could limit the method's practicality. It would be beneficial if the authors could provide further justification for this design choice, perhaps through an ablation study on its precise contribution to performance versus its computational cost.

Overall, while these components demonstrate innovativeness, the motivation behind their design and enablement remains insufficiently articulated. The authors are strongly encouraged to provide more in-depth explanations regarding these aspects.

---

> ### Author Rebuttal · Authors · 2025-07-31
>
> Dear Reviewer Bnjx,
>
> We sincerely appreciate your recognition of our framework’s structural innovation and logical design, and we’re grateful you acknowledged the originality of our multi-expert integration. Your thoughtful examination of the adapter design and multi-expert distillation has strengthened our presentation. We've prepared detailed explanations below addressing your workflow description and component motivation questions.
>
> > **W1: Research on FL mechanisms**
>
> In most FL research, weighted averaging of model parameters or weighted averaging of prototypes are common techniques for integration and processing. To address the unique challenge of data heterogeneity in FL, we enable knowledge sharing among clients by aggregating multi-scale prototypes, while addressing task heterogeneity through internal expert collaboration. Furthermore, we propose **uncertainty-weighted multi-teacher self-distillation**, which dynamically adjusts knowledge transfer weights based on the entropy of expert outputs, avoiding the limitations of simple prototype averaging.
>
> Additionally, in our experiments, we address the core issue of **client data distributions evolving over time** and being **cross-task non-IID**. We simulate unique FL scenarios using two data partitioning strategies, and validate MultiFCL's performance under extreme heterogeneity conditions. In the future, we will explore more FL-specific mechanisms based on your suggestions.
>
> > **W2: Figure 1 modification**
>
> Thank you for your correction. We will further optimize Figure 1 to fully illustrate the workflow of MultiFCL and improve readability.
>
> > **W3: Component workflow**
>
> Below is our supplementary description for these four components:
>
> **Multi-Adapter Fine-Tuning:**
>
> - **Functionality:** Inserts lightweight adapters into each layer of the PTM, fine-tuning them locally only on the first task. The adapters enable the PTM to rapidly adapt to downstream tasks, and are frozen for subsequent tasks to prevent catastrophic forgetting.
>
> - **Connection:** Serves as the PTM fine-tuning module for the entire framework. Its output features are reused by all subsequent components. The freezing mechanism preserves the PTM's generalization capability, providing stable representations for subsequent cross-task knowledge fusion.
>
> **Multi-Modal Prototype Initialization:**
>
> - **Functionality:** Selects the most relevant old-class semantic features and fuses them with the features of the new class sample using cross-attention to initialize the new class prototype. This process incorporates implicit bias from historical tasks, enhancing the discriminative power of the new class representation.
>
> - **Connection:** Relies on features extracted by the PTM after adapter fine-tuning. Injects knowledge from old tasks into new class initialization, avoiding old knowledge forgetting caused by random initialization. Works together with the adapter freezing mechanism to ensure model stability.
>
> **Multi-Scale Feature Learning:**
>
> - **Functionality:** Divides the PTM into $M$ modules, with each module outputting features at different scales. Enforces experts to pull sample features closer to their corresponding class prototypes. Clients upload locally computed multi-scale prototypes to the server for aggregation, forming global class prototypes used in the next round of training.
>
> - **Connection:** Input relies on features extracted by the PTM after adapter fine-tuning. Class prototype updates are based on the multi-modal initialization results. The aggregated prototypes from the server fuse hierarchical features from different clients and are downloaded back to clients for use in multi-teacher self-distillation training.
>
> **Multi-Teacher Dynamic Self-Distillation:**
>
> - **Functionality:** Treats the first $M-1$ experts as teachers. Introduces an uncertainty weighting mechanism that dynamically adjusts distillation weights, suppressing interference from high-uncertainty experts and enhancing the contribution of low-uncertainty experts.
>
> - **Connection:** Takes the outputs of the experts from multi-scale feature learning as input. Solves inference efficiency issues through knowledge compression. The dynamic weighting mechanism prioritizes fusing deep, abstract knowledge, avoiding the influence of shallow noise, further enhancing the discriminative power of the final expert.
>
> The overall workflow can be summarized as:
>
> - **Phase 1: Task Initialization**: Clients train adapters on the first task and upload them to the server for aggregation. Aggregated adapters are distributed back to clients and frozen.
>
> - **Phase 2: New Task Processing**: Clients extract features using the frozen adapters, initialize new class prototypes using semantic features from old tasks. Multi-scale experts are trained using extracted multi-scale features for multi-teacher self-distillation. After local training, clients upload their locally computed prototypes to the server for aggregation.
>
> - **Phase 3: Continual Learning Loop**: Global prototypes are fed back to clients to guide learning on new tasks. This forms a closed loop: Local Training $\to$ Prototype Aggregation $\to$ Knowledge Feedback.
>
> > **W4: Adapter fine-tuning**
>
> We will supplement references related to LoRA. Although similar, the fine-tuning parameters of adapter and LoRA are different. Adapter introduces small neural network modules *independently* into various layers of the model, separate from its main structure. Only these adapter parameters are updated during fine-tuning. In contrast, LoRA introduces low-rank matrices into the model's weight matrices. These low-rank matrices act as modifications applied to the original weight matrices, adjusting them during computation.
>
> Regarding the issue of adapter freezing, as demonstrated in Figure 6 of the paper (Appendix A.8), continuously training the adapters leads to a rapid and severe decline in accuracy. This is because the PTM lies in its generalizable features acquired through large-scale pre-training. Fully fine-tuning the PTM causes the learned feature space to shift towards new tasks, damaging performance on old tasks and resulting in catastrophic forgetting.
>
> Although freezing the adapter might potentially limit cross-domain adaptation capability, this challenge is effectively addressed through the synergistic combination of **multi-modal prototype initialization** and **multi-scale expert distillation**. Specifically, we leverage semantic features to generate the initialization values for new class prototypes through cross-attention mechanisms, thereby transferring abstract knowledge from old domains to new ones. Concurrently, the PTM is partitioned into multi-level expert modules. Uncertainty-weighted dynamic distillation facilitates the reorganization of features across these experts, and the final expert integrates knowledge from all scales. This coordinated approach significantly enhances both generalization capability under domain shifts and task-specific adaptability.
>
> > **W5: Multi-modal knowledge introduction**
>
> The textual information used in the paper to initialize new class prototypes is derived directly from the PTM’s own text encoder. It generates semantic features of old-task class labels through this encoder, without introducing any additional external corpora or knowledge bases. This ensures **the text prototypes solely carry category semantic information directly relevant to the visual task**.
>
> Additionally, we conduct ablation experiments by removing multi-modal initialization (`w/o init.`), as follows:
>
> ||6|4|2|0.5|0.1|0.05|
> |-|-|-|-|-|-|-|
> |CLIP|64.3|63.6|62.7|65.0|63.3|60.3|
> |MFCL|30.4|24.6|15.9|32.9|31.4|30.7|
> |LANDER|20.3|19.8|18.7|34.8|33.4|32.1|
> |PiLora|70.5|63.8|63.6|69.6|63.8|63.0|
> |MultiFCL-w/o init.|69.7|68.9|68.2|69.3|67.6|65.8|
> |MultiFCL|71.0|70.3|69.7|70.7|68.9|67.9|
>
> Table 1: Comparison results of last accuracy on CIFAR100 with 10 tasks in both quality-based partitioning and distribution-based partitioning.
>
> It can be observed that although removing multi-modal initialization leads to performance degradation, the interaction of other components in MultiFCL still enables the model to maintain strong performance, consistently outperforming other methods.
>
> > **W6: Multi-scale feature learning**
>
> Each divided module of the model acts as an independent expert, outputting multi-scale prototypes and **does not perform classification tasks**. Classification tasks are performed **by the final layer expert** after the model is trained via multi-teacher self-distillation. Simultaneously, class prototypes are obtained through iterative updates (Eq. 5) rather than repeated training, with the computation involving only the weighted averaging of feature vectors.
>
> Furthermore, MultiFCL does not impose restrictions on the number of experts established. In our experiments, we set up 4 experts, and this computational overhead is manageable. In scenarios with extreme resource constraints, clients can choose an appropriate number of experts by balancing performance and computational resources.
>
> > **W7: Multi-teacher dynamic self-distillation**
>
> As shown in the ablation experiments of the paper (Tables 3 and 6), after introducing this component, the model achieves the highest accuracy across all data partitioning settings on CIFAR100. This proves its effectiveness in consolidating knowledge from multiple experts and improving final classification performance.
>
> Regarding computational overhead, multi-teacher self-distillation can reduce the number of experts that need to be computed during testing. As described in W6, this avoids the need for every expert to perform classification tasks. During the testing phase, **only the final single-expert model needs to be executed, significantly reducing inference overhead**. Overall, this design achieves a balance between inference efficiency and classification performance through knowledge consolidation during the training phase.

---

> > ### Comment · Reviewer_Bnjx · 2025-08-05
> >
> > Thank you for your detailed and comprehensive rebuttal. After carefully reviewing your responses, most of my previous concerns have been addressed. However, I still have some concerns regarding the computational resources of this method, and therefore, I have decided to maintain my favorable rating.

---

> > > ### Author Response · Authors · 2025-08-05
> > > **Thanks for your reply**
> > >
> > > We appreciate your carefully reviewing our responses. As for computational resources, we compare wall-clock per round for MultiFCL, PiLoRA, and LANDER on CIFAR100. Each round includes 5 local epochs across 10 clients plus one server update. And the three methods have similar GPU memory occupation under the same batch size. The results are as follows:
> > >
> > > || Wall-clock|
> > > |-|-|
> > > |LANDER|169.65s|
> > > |PiLoRA|207.10s|
> > > |MultiFCL|228.29s/150.24s|
> > >
> > > Table 2: Comparison results of wall-clock per round on CIFAR100 with 10 tasks in distribution-based partitioning $\beta=0.05$.
> > >
> > > LANDER integrates pre-trained CLIP and generator yields wall-clock per round comparable to PTM-based methods while requiring 100 rounds per task. PiLoRA continuous LoRA tuning prevents effective knowledge retention while increases computation time. MultiFCL freezes adapters after initial tasks, reducing wall-clock per round (228.29s to 150.24s) while achieving competitive performance.
> > >
> > > We hope the above response can solve your concerns. Once again, thanks so much for your time and efforts engaged in our paper.

---

### Official Review · Reviewer_3UFP · 2025-07-03

**Clarity:** 3
**Significance:** 3
**Originality:** 3
**Rating:** 4
**Confidence:** 4

**Summary:**

This paper proposes MultiFCL, a federated continual learning framework leveraging pre-trained models such as CLIP with ViT backbones. MultiFCL introduces a combination of lightweight adapters, multi-modal prototype initialization, multi-scale feature learning, and multi-teacher dynamic self-distillation to balance stability (preventing catastrophic forgetting) and plasticity (adapting to new tasks) in continual learning. The authors conduct extensive experiments on CIFAR100 and CUB-200-2011 under various partitioning and few-shot scenarios, demonstrating performance improvements over several recent baselines.

**Questions:**

1. The authors mention using TinyImageNet and ImageNet-R in your experimental setup (Section 4.1), but I could not locate any quantitative results. Could you clarify whether you conducted experiments on these datasets and include the results?

2. All experiments appear to rely on CLIP ViT. Have the authors tested MultiFCL with other pre-trained models (e.g., standard ViT models, ResNet-based PTMs) to validate its general applicability?

3. To what extent does MultiFCL’s success depend on the quality of the underlying PTM? Could the authors provide insights or experiments showing performance with weaker PTMs?

**Ethical Concerns:**

["NO or VERY MINOR ethics concerns only"]

**Final Justification:**

The authors provided the detailed response with additional experiments. I keep my initial positive rating.

**Limitations:**

The paper provided a discussion with limitations.

**Paper Formatting Concerns:**

No major formatting issues found.

**Quality:**

3

**Strengths And Weaknesses:**

Strengths
+ The proposed method looks novel. The combination of adapters frozen after the first task, multi-scale experts, and dynamic self-distillation is well-motivated and original.
+ The paper is clearly written overall, with good diagrams and thorough ablations.
+ The experiments are extensive and cover multiple data heterogeneity scenarios, few-shot learning, and ablation studies.

Weaknesses
- Dependency on CLIP ViT: The approach heavily relies on CLIP as the backbone. It is unclear how well the method generalizes to other architectures or less powerful pre-trained models.
- The authors state that they use TinyImageNet and ImageNet-R for evaluation (Sec 4.1), but I could not find any reported results for these datasets in the main text or appendix, even though they showed some heterogeneity study on ImageNet-R dataset. This omission weakens the empirical validation.
- While the experiments are extensive, they only explore a single PTM architecture (CLIP ViT). Evaluating with other PTMs or even smaller ViT models would strengthen claims of generalizability.
- Some implementation details, for example, adapter size, hyperparameter settings for all datasets, are scattered or only briefly mentioned.

---

> ### Author Rebuttal · Authors · 2025-07-31
>
> Dear Reviewer 3UFP,
>
> We're honored by your praise for our method's originality and the comprehensive ablation studies. Your perceptive questions about model applicability have significantly broadened our perspective. In the responses below, we've included additional experimental insights from TinyImageNet/ImageNet-R evaluations and different PTM architectures to clarify your technical inquiries.
>
> > **W1 (Q2): All experiments appear to rely on CLIP ViT. Have the authors tested MultiFCL with other pre-trained models (e.g., standard ViT models, ResNet-based PTMs) to validate its general applicability?**
>
> MultiFCL relies solely on CLIP’s image-text alignment capability and can be extended to any PTM with such alignment. For example, we supplement Flava [1] as our backbone model and report its last accuracy results on CIFAR100 below:
>
> [1] Amanpreet Singh et al., FLAVA: A Foundational Language And Vision Alignment Model
>
> ||6|4|2|0.5|0.1|0.05|
> |-|-|-|-|-|-|-|
> |CLIP-based|71.0|70.3|69.7|70.7|68.9|67.9|
> |Flava-based|70.4|69.9|69.7|69.5|68.2|67.5|
>
> Table 1: Comparison results of different backbone models on CIFAR100 with 10 tasks in both quality-based partitioning and distribution-based partitioning.
>
> As shown in Table 1, MultiFCL **maintains relatively stable performance across different PTMs**. Beyond this, image-text alignment can also be achieved by adding projection layers, offering a pathway to extend compatibility with more PTMs. In the future, we will explore novel architectures to enhance MultiFCL’s scalability.
>
> > **W2 (Q1): The authors mention using TinyImageNet and ImageNet-R in your experimental setup (Section 4.1), but I could not locate any quantitative results. Could you clarify whether you conducted experiments on these datasets and include the results?**
>
> Figure 4 in the paper (Appendix A.5) shows the experimental results of MultiFCL on CIFAR100, CUB-200-2011, TinyImageNet, and ImageNet-R under six different data partitions and two distinct task partitioning strategies. These results include detailed performance of MultiFCL across incremental tasks. Additionally, as per your suggestion, we supplement the experiments with PiLoRA, CLIP on TinyImageNet and ImageNet-R. The results are as follows:
>
> ||6|4|2|0.5|0.1|0.05|
> |-|-|-|-|-|-|-|
> |CLIP|57.6|57.4|57.4|57.9|58.0|57.3|
> |PiLora|64.4|62.2|61.8|61.9|60.6|58.2|
> |MultiFCL|**66.1**|**65.0**|**64.8**|**64.4**|**63.0**|**61.6**|
>
> Table 2: Comparison results of last accuracy on TinyImageNet with 10 tasks in both quality-based partitioning and distribution-based partitioning.
>
> ||6|4|2|0.5|0.1|0.05|
> |-|-|-|-|-|-|-|
> |CLIP|58.7|58.2|58.1|59.9|59.3|58.4|
> |PiLora|63.2|62.6|61.2|67.7|67.8|66.9|
> |MultiFCL|**69.3**|**67.7**|**65.4**|**74.0**|**73.5**|**72.9**|
>
> Table 3: Comparison results of last accuracy on ImageNet-R with 10 tasks in both quality-based partitioning and distribution-based partitioning.
>
> It can be observed that MultiFCL demonstrates impressive results across these datasets.
>
> > **W3 (Q3): To what extent does MultiFCL’s success depend on the quality of the underlying PTM? Could the authors provide insights or experiments showing performance with weaker PTMs?**
>
> Following your suggestion, we supplement the results using TinyCLIP ViT-39M/16 Text-19M [2] on CIFAR100. This is a distilled version of CLIP with significantly fewer pre-training parameters than CLIP. The results are as follows:
>
> [2] Kan Wu et al., TinyCLIP: CLIP Distillation via Affinity Mimicking and Weight Inheritance
>
> ||6|4|2|0.5|0.1|0.05|
> |-|-|-|-|-|-|-|
> |CLIP-based|71.0|70.3|69.7|70.7|68.9|67.9|
> |TinyCLIP ViT-39M-based|70.8|70.2|69.5|70.9|70.0|69.2|
>
> Table 3: Comparison results of last accuracy on CIFAR100 with 10 tasks in both quality-based partitioning and distribution-based partitioning.
>
> It can be seen that using TinyCLIP does not degrade MultiFCL's performance. Conversely, it remains highly stable, which demonstrates **the potential for balancing resource constraints and performance requirements in resource-limited scenarios** (e.g., edge devices). Moreover, we note that multiple distilled versions of TinyCLIP exist. In the future, we will experiment with other variants to further assess MultiFCL’s adaptability across diverse model scales.
>
> > **W4: Some implementation details, for example, adapter size, hyperparameter settings for all datasets, are scattered or only briefly mentioned.**
>
> In response to your suggestions, we supplement specific details regarding the adapter size: An adapter consists of $W_{down}=768\times 8+8$ parameters and $W_{up}=768\times 8+768$ parameters, totaling 13,064 parameters. Additionally, the positions mentioning hyper-parameters for all datasets and other experimental settings will be adjusted to ensure they are consolidated together.

---

> > ### Comment · Reviewer_3UFP · 2025-08-04
> >
> > I would like to thank the authors for the detailed response with the additional experimental results. I keep my initial positive rating.

---

> > > ### Author Response · Authors · 2025-08-04
> > > **Thanks for your reply**
> > >
> > > Thanks so much for your time and efforts engaged in our paper. We are grateful for your valuable suggestions.

---

### Official Review · Reviewer_6opu · 2025-07-03

**Clarity:** 3
**Significance:** 3
**Originality:** 3
**Rating:** 4
**Confidence:** 4

**Summary:**

In federated continual learning (FCL), achieving both stability (to preserve performance on old tasks) and plasticity (to learn new tasks effectively) is important. This paper uses pre-trained models (like CLIP/ViT) with lightweight adaptations to improve both stability and plasticity in continual federated settings. It first introduces adapters to quickly adapt PTMs to downstream tasks, which are then frozen to prevent catastrophic forgetting. Then multi-modal prototype initialization, multi-scale feature learning, multi-teacher dynamic self-distillation strategies have been proposed to effectively preserve the knowledge of prior tasks while learning new knowledge in federated continual learning. Experiment is conducted using CIFAR10, CIFAR100, TinyImageNet datasets on CLIP model.

**Questions:**

Please see the weakness 1, 2, 3 above, and try to address the issues.

In addition to the above weaknesses, I believe the paper could benefit from providing the overall pseudo code of the algorithm. It's difficult to know when model aggregation is conducted, when specific losses are used locally, etc.

**Ethical Concerns:**

["NO or VERY MINOR ethics concerns only"]

**Final Justification:**

The authors have addressed my concerns, including results with more clients, communication/computation, and pseudo code, better clarifications. The authors also have addressed other reviewers' comments. I am leaning towards acceptance for this paper, so I keep my original score borderline accept.

**Limitations:**

Yes the limitation is well addressed.

**Quality:**

3

**Strengths And Weaknesses:**

Strength

1. This paper appropriately integrates new techniques to handle the continual federated learning problem. Multi-modal prototype initialization, multi-scale feature learning, multi-teacher dynamic self-distillation strategies are relatively novel to tackle this problem.

2. Experiments are conducted using various datasets and the effect of each component, including self distillation, multi-modal prototype initialization, have been well studied.

Weakness

1. In practice, there could be many clients in the system but the paper considers only 10 clients.

2. The intuition behind each component is less illustrated. For example, the adapters are only updated based on the first task and is frozen. So this means that this approach can preserve the knowledge of the first task, but not the knowledge of subsequent tasks. So how are the knowledge of subsequent tasks are preserved? In general I feel that the advantage/effect of each scheme should be better illustrated instead of just describing how they work.

3. The computational/communication costs are analyzed, but they are not compared with the ones of the baselines. How much additional burden does this approach require compared to baselines?

---

> ### Author Rebuttal · Authors · 2025-07-31
>
> Dear Reviewer 6opu,
>
> Thank you for acknowledging our novel integration of multi-modal prototypes and dynamic distillation. We're also particularly grateful your attention to the thorough component ablation studies. Your perspective on client scalability, the intuition behind each component, computational trade-offs, and pseudo code display has enriched our discussion. Detailed responses to your questions about them are provided below.
>
> > **W1: In practice, there could be many clients in the system but the paper considers only 10 clients.**
>
> Per your suggestion, we investigate the last-task accuracy of MultiFCL, PiLoRA, and CLIP under varying the number of clients on CIFAR100. The results are as follows:
>
> ||10|30|50|70|90|
> |-|-|-|-|-|-|
> |CLIP|60.3|59.8|59.9|60.1|60.0|
> |PiLora|63.0|63.6|63.2|63.7|63.3|
> |MultiFCL|**67.9**|**67.9**|**66.9**|**68.1**|**67.9**|
>
> Table 1: Comparison results of different number of clients on CIFAR100 with 10 tasks in distribution-based partitioning $\beta=0.05$.
>
> With fixed dataset size and numerous incremental tasks, increasing the number of clients reduces per-client data size and intensifies extreme heterogeneity in data distribution across clients. Nevertheless, MultiFCL consistently **maintains superior performance versus other methods, demonstrating its robustness**.
>
> > **W2: The intuition behind each component is less illustrated. For example, the adapters are only updated based on the first task and is frozen. So this means that this approach can preserve the knowledge of the first task, but not the knowledge of subsequent tasks. So how are the knowledge of subsequent tasks are preserved? In general I feel that the advantage/effect of each scheme should be better illustrated instead of just describing how they work.**
>
> Based on your suggestion, we provide the following supplementary explanation for the intuition behind each component:
>
> - **Multi-Adapter Fine-Tuning**:
>
> The core objective of the adapters is to **bridge the domain gap** between the PTM and downstream tasks while avoiding catastrophic forgetting. Traditional full-parameter fine-tuning would disrupt the PTM's **generalization capabilities**, whereas completely freezing parameters prevents adaptation to new tasks. Therefore, by fine-tuning lightweight adapters on the first task and then freezing them, we lock the PTM's adaptation capability for that initial task. This prevents subsequent task updates from overwriting learned knowledge. Knowledge retention for subsequent tasks no longer relies on adapter updates but is instead achieved through later components (e.g., prototype initialization), **thus preventing generalization degradation caused by excessive fine-tuning**.
>
> - **Multi-Modal Prototype Initialization**:
>
> This component addresses the **initialization bias** problem for new class prototypes in FCL. Traditional random initialization causes the new class classifier to lack historical task knowledge, leading to confusion between new and old classes. This component enables the fusion of features from new class samples with semantic features from similar old classes to initialize the new class prototype. Its essence is to **introduce semantic priors from old tasks**, positioning the initial location of the new class prototype near the feature space of semantically related old classes, which **enhances inter-class discriminability and reduces forgetting**.
>
> - **Multi-Scale Feature Learning**:
>
> To address the insufficient plasticity of the model under dynamic data distributions, we partition the PTM into multi-level experts. Shallow experts extract textural features, while deep experts capture semantic features, collectively providing diverse representations of samples. Each expert independently optimizes its class prototypes, driving samples of the same class towards a unified class center within their respective feature spaces. Simultaneously, aggregating multi-scale prototypes across clients forms global class centers, addressing the feature shift problem caused by data heterogeneity. This multi-expert mechanism allows clients to learn new tasks at different granularities while **implicitly fusing historical task knowledge through prototype sharing**.
>
> - **Multi-Teacher Dynamic Self-Distillation**:
>
> To balance **inference efficiency** and **knowledge integration** within the multi-expert system, we distill knowledge from all experts into the final expert, reducing computational overhead during testing. Since features from deeper experts are generally more reliable, we design an uncertainty weighting scheme. This down-weights contributions from shallow experts with high uncertainty to **prevent them from interfering with the final decision**. This component maintains multi-scale information while enhancing inference efficiency. The weighting mechanism also strengthens the transfer of reliable knowledge.
>
> > **W3: The computational/communication costs are analyzed, but they are not compared with the ones of the baselines. How much additional burden does this approach require compared to baselines?**
>
> We compare communication parameters and wall-clock per round for MultiFCL, PiLoRA, MFCL, and LANDER on CIFAR100. Each round includes 5 local epochs across 10 clients plus one server update. The results are as follows:
>
> ||Communication|Wall-clock|
> |-|-|-|
> |MFCL|24.27M|35.20s|
> |LANDER|11.20M|169.65s|
> |PiLoRA|0.0584M|207.10s|
> |MultiFCL|0.54/0.38M|228.29s/150.24s|
>
> Table 2: Comparison results of communication parameters and wall-clock per round on CIFAR100 with 10 tasks in distribution-based partitioning.
>
> In MFCL, the server must distribute three components to clients, including the current-task global model, previous-task global model, and generator. Although the global model uses ResNet-18 (11.2M), this still incurs substantial communication overhead. Meanwhile, despite its wall-clock per round remains low, it requires 100 communication rounds per task to achieve competitive results, whereas MultiFCL and PiLoRA attain expressive outcomes within merely 5 rounds.
>
> LANDER transmits the ResNet-18 as global model each round. Its integration of pre-trained CLIP and generator yields wall-clock per round comparable to PTM-based methods while requiring 100 rounds per task.
>
> PiLoRA communicates LoRA parameters and prototypes per round. Although parameter volume stays low, continuous LoRA tuning prevents effective knowledge retention and increases computation time, resulting in suboptimal performance versus MultiFCL.
>
> MultiFCL freezes adapters after initial tasks, reducing communication parameters (0.54M to 0.38M) and wall-clock per round (228.29s to 150.24s) while achieving competitive performance within 5 communication rounds per task. Collectively, MultiFCL **maintains computational efficiency while reducing communication costs versus existing methods**.
>
> > **Q1: I believe the paper could benefit from providing the overall pseudo code of the algorithm. It's difficult to know when model aggregation is conducted, when specific losses are used locally, etc.**
>
> According to your suggestion, the following is the pseudo code of MultiFCL. We will update the more detailed version to the paper:
>
> **Require**: Pre‑trained model $f$, clients $K$, communication rounds $T$, tasks $\mathcal{T}_{\tau}$, local epochs $E$
>
> > **for** each task:
> > &nbsp;&nbsp;**if** t = 1:
> > &nbsp;&nbsp;&nbsp;&nbsp;**for** round $r = 1$ **to** $T$:
> > &nbsp;&nbsp;&nbsp;&nbsp;&nbsp;&nbsp;**for** each client $k = 1$ **to** $K$ *(in parallel)*:
> > &nbsp;&nbsp;&nbsp;&nbsp;&nbsp;&nbsp;&nbsp;&nbsp;Fine‑tune adapters on $D_k^1$
> > &nbsp;&nbsp;&nbsp;&nbsp;&nbsp;&nbsp;&nbsp;&nbsp;Upload $w_k^r$ and sample count $|X_k^1|$ to server
> > &nbsp;&nbsp;&nbsp;&nbsp;**Server aggregates**:
> > &nbsp;&nbsp;&nbsp;&nbsp;&nbsp;&nbsp;$w^r \leftarrow \sum_{k=1}^K \frac{|X_k^1|}{\sum |X_k^1|} w_k^r$
> > &nbsp;&nbsp;&nbsp;&nbsp;**Server broadcasts** global adapter $w^r$ to clients
> > &nbsp;&nbsp;Freeze global adapters: $w^* \leftarrow w^T$
> > &nbsp;&nbsp;**else**:
> > &nbsp;&nbsp;&nbsp;&nbsp;**for** each new class $c$ in new task:
> > &nbsp;&nbsp;&nbsp;&nbsp;&nbsp;&nbsp;Compute semantic similarity
> > &nbsp;&nbsp;&nbsp;&nbsp;&nbsp;&nbsp;Initialize prototype $\mathbf{p}_{\mathrm{init}}^c$
> >
> > &nbsp;&nbsp;**for** round $r = 1$ **to** $T$:
> > &nbsp;&nbsp;&nbsp;&nbsp;**for** each client $k = 1$ **to** $K$ *(in parallel)*:
> > &nbsp;&nbsp;&nbsp;&nbsp;&nbsp;&nbsp;**for** local epoch $e = 1$ **to** $E$:
> > &nbsp;&nbsp;&nbsp;&nbsp;&nbsp;&nbsp;&nbsp;&nbsp;Extract multi‑scale features
> > &nbsp;&nbsp;&nbsp;&nbsp;&nbsp;&nbsp;&nbsp;&nbsp;Compute expert loss
> > &nbsp;&nbsp;&nbsp;&nbsp;&nbsp;&nbsp;&nbsp;&nbsp;Update parameters $\theta \leftarrow \theta - \eta \nabla_\theta (\mathcal{L})$
> > &nbsp;&nbsp;&nbsp;&nbsp;&nbsp;&nbsp;&nbsp;&nbsp;Update prototypes
> > &nbsp;&nbsp;&nbsp;&nbsp;&nbsp;&nbsp;Upload prototypes and sample counts to server
> > &nbsp;&nbsp;**Server aggregates prototypes**:
> > &nbsp;&nbsp;&nbsp;&nbsp;$\mathbf{p}^{c,r} \gets \sum_{k=1}^K \frac{|X_k^{c,t}|}{\sum |X_k^{c,t}|} \sum_m \mathbf{p}_{k,m}^{c,r}$
> > &nbsp;&nbsp;**Server broadcasts** global prototypes $\mathbf{p}^r$ to clients

---

> > ### Comment · Reviewer_6opu · 2025-08-01
> >
> > The authors have addressed my concerns as well as other reviewers' comments. I am leaning towards acceptance for this paper, so I keep my original score borderline accept.

---

### Official Review · Reviewer_aSUp · 2025-07-05

**Clarity:** 3
**Significance:** 3
**Originality:** 3
**Rating:** 4
**Confidence:** 4

**Summary:**

The paper proposes MultiFCL, an FCL framework that (i) trains task-specific adapters on the first task and then freezes them, (ii) initialises class prototypes by fusing CLIP visual embeddings with label text embeddings, and (iii) introduces several “experts” (different depth slices of the ViT backbone) whose predictions are merged into a single head via uncertainty-weighted self-distillation.

**Questions:**

1. Can you report true communication (MB) and wall-clock per round on a standard GPU cluster? Currently efficiency claims are speculative.

1. Prototype sharing may leak class information. Can you provide an empirical or theoretical privacy analysis, or add differential-privacy noise?

**Ethical Concerns:**

["NO or VERY MINOR ethics concerns only"]

**Limitations:**

yes

**Quality:**

2

**Strengths And Weaknesses:**

Strengths
1. Implementation details (optimiser, epochs, learning rates) are mostly disclosed and code link is promised.
1. High-level intuition is understandable.
1. Shows that freezing CLIP with adapters can outperform training-from-scratch FCL baselines.

Weaknesses
1. No statistical significance: all results are single runs, no error bars.
1. Communication/time claims lack real wall-clock or FLOP evidence.
1. Gains largely attributable to CLIP pre-training; incremental benefit of the expert/distillation machinery is small.
1.Datasets are moderate-scale; no real FL use-case (health, IoT) included.

---

> ### Author Rebuttal · Authors · 2025-07-31
>
> Dear Reviewer aSUp,
>
> We sincerely appreciate your recognition of our framework's high-level intuition and the effectiveness of adapter freezing. Your insights on statistical significance and communication efficiency have helped us deepen our analysis. We've carefully addressed your technical questions about empirical validation and privacy protection in the following responses, which we hope will resolve your concerns.
>
> > **W1: No statistical significance: all results are single runs, no error bars.**
>
> Following your suggestion, we conduct experiments on CIFAR100 under distribution-based partitioning using five distinct random seeds. We supplement standard deviation for the last-task accuracy of MultiFCL, PiLoRA, and MFCL-CLIP. The results are as follows:
>
> ||0.5|0.1|0.05|
> |-|-|-|-|
> |MFCL-CLIP|52.3±0.4|51.8±0.4|50.6±1.0|
> |PiLoRA|69.2±0.4|63.2±0.5|62.7±0.5|
> |MultiFCL|**70.7±0.3**|**68.7±0.2**|**67.7±0.2**|
>
> Table 1: Comparison results of last accuracy on CIFAR100 with 10 tasks in distribution-based partitioning $\beta=0.05$.
>
> The p-values for MultiFCL versus PiLoRA and MultiFCL versus MFCL-CLIP are both substantially below 0.001. This demonstrates that MultiFCL **achieves stable performance with statistically significant improvements**.
>
> > **W2 (Q1): Communication/time claims lack real wall-clock or FLOP evidence.**
>
> We compare communication parameters and wall-clock per round for MultiFCL, PiLoRA, MFCL, and LANDER on CIFAR100. Each round includes 5 local epochs across 10 clients plus one server update. The results are as follows:
>
> ||Communication|Wall-clock|
> |-|-|-|
> |MFCL|24.27M|35.20s|
> |LANDER|11.20M|169.65s|
> |PiLoRA|0.0584M|207.10s|
> |MultiFCL|0.54/0.38M|228.29s/150.24s|
>
> Table 2: Comparison results of communication parameters and wall-clock per round on CIFAR100 with 10 tasks in distribution-based partitioning.
>
> In MFCL, the server must distribute three components to clients, including the current-task global model, previous-task global model, and generator. Although the global model uses ResNet-18 (11.2M), this still incurs substantial communication overhead. Meanwhile, despite its wall-clock per round remains low, it requires 100 communication rounds per task to achieve competitive results, whereas MultiFCL and PiLoRA attain expressive outcomes within merely 5 rounds.
>
> LANDER transmits the ResNet-18 as global model each round. Its integration of pre-trained CLIP and generator yields wall-clock per round comparable to PTM-based methods while requiring 100 rounds per task.
>
> PiLoRA communicates LoRA parameters and prototypes per round. Although parameter volume stays low, continuous LoRA tuning prevents effective knowledge retention and increases computation time, resulting in suboptimal performance versus MultiFCL.
>
> MultiFCL freezes adapters after initial tasks, reducing communication parameters (0.54M to 0.38M) and wall-clock per round (228.29s to 150.24s) while achieving competitive performance within 5 communication rounds per task. Collectively, MultiFCL **maintains computational efficiency while reducing communication costs versus existing methods**.
>
> > **W3: Gains largely attributable to CLIP pre-training; incremental benefit of the expert/distillation machinery is small. Datasets are moderate-scale; no real FL use-case (health, IoT) included.**
>
> In Table 1 of our paper, we present results for ViT, CLIP, MFCL-CLIP, FedMGP, PiLoRA, and PiLoRA-CLIP, which are all PTM-based methods. Among them, MultiFCL consistently outperforms these approaches across all CIFAR100 partitioning configurations, achieving 3.8\%-19.5\% higher average last-task accuracy. Simultaneously, Table 3 of our paper demonstrates MultiFCL delivers 6.4\% higher average accuracy than directly using pre-trained CLIP, validating the efficacy of MultiFCL’s components.
>
> Additionally, following your suggestion, we supplement comparative experiments of MultiFCL, PiLoRA, and CLIP on MedMNIST [1] and IP102 [2]. MedMNIST constitutes a large-scale 2D and 3D medical image dataset incorporating multiple modalities (X-ray images, retinal OCT, ultrasound, CT scans), while IP102 is an agricultural pest/disease dataset exhibiting high inter-class similarity, significant intra-class variance (e.g., identical pests at different life stages), and imbalanced distribution. The results are as follows:
>
> [1] MedMNIST: https://medmnist.com
>
> [2] IP102: https://github.com/xpwu95/IP102
>
> ||6|4|2|0.5|0.1|0.05|
> |-|-|-|-|-|-|-|
> |CLIP|31.2|30.9|20.3|28.4|24.3|21.5|
> |PiLora|33.4|32.5|22.9|30.0|25.3|21.8|
> |MultiFCL|**40.3**|**40.0**|**34.1**|**43.8**|**42.2**|**43.2**|
>
> Table 3: Comparison results of last accuracy on MedMNIST with 10 tasks in both quality-based partitioning and distribution-based partitioning.
>
> ||6|4|2|0.5|0.1|0.05|
> |-|-|-|-|-|-|-|
> |CLIP|35.3|34.6|33.9|40.6|38.9|37.2|
> |PiLora|36.4|35.6|35.3|40.3|38.3|36.5|
> |MultiFCL|**44.9**|**43.8**|**42.6**|**44.8**|**43.8**|**42.5**|
>
> Table 4: Comparison results of last accuracy on IP102 with 10 tasks in both quality-based partitioning and distribution-based partitioning.
>
> It can be seen that MultiFCL maintains robust adaptability compared with other methods in these FL application fields.
>
> **Q2: Prototype sharing may leak class information. Can you provide an empirical or theoretical privacy analysis, or add differential-privacy noise?**
>
> The prototype features are ultimately obtained through mean aggregation, a process that is irreversible. This can be regarded as a linear transformation of the data. Due to the nonlinearity of the feature extraction function, the features of a single sample cannot be separated from the mean. Furthermore, we perform feature inversion on both the communicated prototypes and the features extracted from individual image samples. We observe that these prototypes **cannot reconstruct the approximate outlines or sensitive information of the original images**. Only the features extracted from a single sample are able to reconstruct sensitive information from the original image. In the future, we will explore adding Gaussian noise to the communicated prototypes to enhance privacy protection, as you suggested, which may require balancing privacy protection against model performance.

---

### Note · Authors · 2025-08-11

We sincerely appreciate the recognition from all reviewers regarding the innovative contributions of this paper. We have carefully considered each comment and provided detailed responses in the rebuttal, along with necessary clarifications and supplementary experiments. Below is a summary of the key points:

> **Reviewers without follow-up feedback**

Although Reviewer aSUp did not respond, we addressed his concerns through additional experiments: **multi-round random seed experiments and statistical tests** validated the significance of MultiFCL's performance improvements, **empirical measurements of communication costs and wall-clock time** demonstrated MultiFCL's efficiency, cross-domain experimental results on **medical imaging and agricultural pest/disease datasets** proved MultiFCL's superiority in real-world scenarios, **feature inversion analysis** clarified the non-invertibility of prototypes, providing a basis for privacy protection.

> **Reviewers with follow-up feedback**

We are pleased that the remaining reviewers confirmed their main concerns were resolved and provided positive scores of MultiFCL:

For Reviewer 6opu and Reviewer 3UFP, we confirmed model stability through client scalability experiments, new explanations of component motivations and algorithmic pseudo-code clarified the multi-expert collaboration mechanism, supplementary results on TinyImageNet/ImageNet-R and experiments with Flava/TinyCLIP further verified MultiFCL's versatility.

For Reviewer Bnjx, we provided in-depth explanations of FL mechanisms and component synergy, particularly validating dynamic distillation's contribution via ablation studies. Regarding computational efficiency, we emphasized that the multi-expert design only **extracts hierarchical feature prototypes**, prototype updates employ **lightweight vector weighting**, and the number of experts can be flexibly adjusted based on edge device resources. Additionally, we supplemented **wall-clock comparisons with baselines** (as recognized by Reviewer 6opu), demonstrating that MultiFCL maintains computational efficiency while achieving competitive performance.

> **Conclusion**

All four reviewers acknowledged MultiFCL's contributions positively. We hope that our rebuttal, featuring new experiments across diverse datasets with various backbones and configurations, along with empirical measurements of communication/computation costs and thorough technical analysis, has resolved all reviewers' concerns.

---

### Decision · Program_Chairs · 2025-09-17

**Decision:**

Accept (poster)

**Comment:**

This paper introduces MultiFCL, a federated continual learning framework that leverages pre-trained models with lightweight adapters, multi-modal prototype initialization, multi-scale feature learning, and multi-teacher dynamic self-distillation to balance stability and plasticity. Reviewers agreed that the method is novel, well-motivated, and supported by extensive experiments across diverse datasets, including federated and cross-domain settings. Concerns were raised about reliance on CLIP backbones, missing efficiency evidence, and unclear component motivations, but the rebuttal provided new experiments, statistical analyses, computational cost measurements, and detailed clarifications that addressed most issues. Some reviewers still noted heavy dependence on PTMs and computational overhead, but these were judged to be acceptable trade-offs given the strong empirical gains. Overall, the paper offers a technically sound and innovative contribution to FCL, with convincing results and thorough ablations